# A regime view of future atmospheric circulation changes in Northern mid-latitudes

Federico Fabiano[1], Virna L. Meccia[1], Paolo Davini[2], Paolo Ghinassi[1], and Susanna Corti[1]

[1]Institute of Atmospheric Sciences and Climate (ISAC-CNR), Bologna, Italy
[2]Institute of Atmospheric Sciences and Climate (ISAC-CNR), Turin, Italy

**Correspondence:** F. Fabiano (f.fabiano@isac.cnr.it)

**Abstract.** Future wintertime atmospheric circulation changes in the Euro-Atlantic (EAT) and Pacific-North American (PAC) sectors are studied from a weather regimes perspective. The CMIP5 and CMIP6 historical simulations performance in reproducing the observed regimes is first evaluated, showing a general improvement of CMIP6 models, more evident for EAT. The circulation changes projected by CMIP5 and CMIP6 scenario simulations are analysed in terms of the change in the frequency and persistence of the regimes. In the EAT sector, significant positive trends are found for the frequency and persistence of NAO+ for SSP2-4.5, SSP3-7.0 and SSP5-8.5 scenarios, with a concomitant decrease in the frequency of the Scandinavian Blocking and Atlantic Ridge regimes. For PAC, the Pacific Trough regime shows a significant increase, while the Bering Ridge is predicted to decrease in all scenarios analysed. The spread among the model responses is linked to different levels of warming in the polar stratosphere, the tropical upper troposphere, the North Atlantic and the Arctic.

## 1 Introduction

A major challenge for the climate community is to understand how a warmer climate will affect the large-scale atmospheric circulation in the mid-latitudes. Indeed, there is growing interest on this topic both from the scientific community and from society, as future changes in circulation are also inextricably related to regional impacts and the occurrence of extreme weather conditions (e.g. Brunner et al., 2018; Schaller et al., 2018; Screen and Simmonds, 2014; Sousa et al., 2018). The wintertime mid-latitude climate in the Northern Hemisphere is primarily influenced by the low-frequency variability (at timescales longer than 5 days) related to the strength and position of the eddy-driven jet stream (Woollings et al., 2010; Barnes and Polvani, 2013). This is particularly true for the North-Atlantic and North-Pacific sectors, where the latitudinal shifts of the jet describe a significant fraction of the low-frequency variability (Athanasiadis et al., 2010) and determine specific impacts locally (Ma et al., 2020) and over downstream regions (i.e. Europe and North America) (Screen and Simmonds, 2014; Zappa et al., 2015a, b; Loikith and Broccoli, 2014). In recent years, several studies on the future evolution of the mid-latitude atmospheric circulation focused on the changes in the mean state of the jet streams, mainly in terms of latitudinal shifts and changes in the jet speed (Barnes and Polvani, 2013). Further attention on the topic has grown in the last decade following the debate about the influence that the Arctic amplification - i.e. the faster warming of the surface at high latitudes - may have on the jet structure (Barnes and Screen, 2015; Hoskins and Woollings, 2015). The emerging picture is that the fate of the eddy-driven jet streams in a

warmer climate is mainly controlled by the meridional temperature gradient in mid-latitudes, which in turn depends on three independent processes, all linked to the differential heating of different regions of the atmosphere:

- the faster warming of the tropical upper troposphere - known as upper tropospheric warming (UTW) -, mainly driven by increased convection and upper-level latent-heat release (Peings et al., 2017). The effect of UTW is to increase the meridional temperature gradient in the upper troposphere and promote a poleward shift and intensification of the jet (Barnes and Screen, 2015).

- The Arctic amplification (AA), which is due primarily to sea-ice retreat and increased heat flux from the ocean in autumn and winter, along with several other positive feedbacks in the Arctic region (Screen and Simmonds, 2010; Pithan and Mauritsen, 2014). The effect of AA is to decrease the low-level meridional temperature gradient and to promote a slow down and equatorward shift of the jet (Peings et al., 2017; Hassanzadeh et al., 2014; Cohen et al., 2019; Overland et al., 2016).

- The change of the polar stratospheric temperature (PST) and consequent feedback on the polar vortex strength (PVS). The fate of the polar stratosphere is still unclear, but there is some indication that it may be of primary importance for the North-Atlantic jet stream (Manzini et al., 2014; Zappa and Shepherd, 2017; Peings et al., 2017).

Due to the large internal variability of the system on interannual to decadal time-scales, detecting circulation changes in the observations has proven to be a challenging task and multiple circulation indices do not show significant trends during the observational period (Blackport and Screen, 2020; Barnes and Screen, 2015). Nevertheless, some robust indications of future changes come from General Circulation Models (GCMs) simulations under greenhouse gases (GHG) forcing scenarios. Analyses on the Coupled Model Intercomparison Project - Phase 5 (CMIP5 Taylor et al., 2012) and Phase 6 (CMIP6 Eyring et al., 2016) ensembles have shown a general agreement for a moderate (about 1 deg) northward shift of the annual-mean zonal-mean eddy-driven jet by 2100 (Barnes and Polvani, 2013; Shaw et al., 2016). However, the picture appears more complex than that, and the jet response strongly depends on the region (Barnes and Polvani, 2013; Peings et al., 2017) and season considered (Barnes and Polvani, 2015; Shaw et al., 2016). Whilst the northward shift of the jet is evident in the North-Pacific (Oudar et al., 2020), the trend over the North-Atlantic shows rather a squeezing of the time-mean jet, with intensification and eastward elongation of the westerlies over Europe (Oudar et al., 2020; Peings et al., 2018). The eastward elongation of the North-Atlantic jet is also consistent with results obtained from analysis of the stationary waves response to climate change, which show an eastward shift in phase, produced by a decrease in the stationary zonal wavenumber (Wills et al., 2019; Simpson et al., 2014). Also from the dynamical point of view, the response in the North-Atlantic appears to be more complex, with concurrent and opposite influences of UTW and AA (Peings et al., 2017). This picture is further complicated by the emerging role of the polar stratosphere, which is strongly coupled with the North-Atlantic jet stream position (e.g. Baldwin and Dunkerton, 2001), and might contribute to the dynamical response to the UTW (Peings et al., 2017; Manzini et al., 2014; Beerli and Grams, 2019), therefore explaining the large inter-model spread (Peings et al., 2017; Zappa and Shepherd, 2017; Oudar et al., 2020).

Besides changes in the mean state, a large interest has been given to changes in the day-to-day variability of the jets, also motivated by the fact that climate extremes are commonly related with persistent circulation anomalies. Barnes and Polvani (2013) showed that, in the North-Atlantic, there is a decreasing trend in the first mode of variability of the jet – related to the latitudinal shifts – and an increase in the second mode – related to variations in jet speed – under RCP8.5 scenario. This picture has been confirmed by the analysis by Peings et al. (2018), which claims that there would be less room for latitudinal shifts of the jet due to the squeezing produced by UTW and AA. Analyses based on various indices of "waviness", i.e. of the departure from a purely zonal jet structure, predict a more zonal flow under a warmer climate (Blackport and Screen, 2020; Peings et al., 2017), with the possible exception of the North-American region (Di Capua and Coumou, 2016; Peings et al., 2017; Vavrus et al., 2017). This is also confirmed by analysis of the atmospheric blocking frequencies in CMIP5 and CMIP6 models, with a general decrease in winter blocking over the Northern Hemisphere and a tendency to an eastward shift of the blocking maxima, with the only small (non-significant) increase over western Canada (Davini and D'Andrea, 2020; Woollings et al., 2018).

In this work, we propose an alternative view of future changes in the wintertime circulation at Northern mid-latitudes, based on the analysis of the daily geopotential height at 500 hPa. With respect to the climatological reference state, midlatitude disturbances appear as transient geopotential height anomalies that can persist beyond the typical synoptic scale, up to three or four weeks. In some regions, the flow tends to organize in some preferred configurations, although the number of such configurations to be considered is still a matter of debate (Hannachi et al., 2017). These preferred large-scale circulation patterns are commonly referred to as weather regimes (WRs) and have been studied mostly for the Euro-Atlantic (EAT: Michelangeli et al., 1995; Dawson et al., 2012; Strommen et al., 2019) and Pacific-North American (PAC: Straus et al., 2007; Weisheimer et al., 2014) sectors. Each WR has a different impact on the climate of the region considered, driving specific precipitation and temperature anomalies. Many works in literature studied how WRs are reproduced by GCMs, but mostly focused on the model performance in control or historical simulations (Dawson et al., 2012; Cattiaux et al., 2013b; Dawson and Palmer, 2015; Weisheimer et al., 2014; Strommen et al., 2019; Fabiano et al., 2020). Changes of WRs in CMIP5 projections were analysed by Cattiaux et al. (2013a) and by Ullmann et al. (2014) for the EAT sector.

From an heuristic perspective, the WRs can be seen as the attractors of a nonlinear dynamical system, whose main characteristics may be described in terms of their position in phase space and their frequency of occurrence. In simple dynamical systems, under a small external forcing the main structure of the attractors in the phase space is only marginally affected by the forcing (at least at the first order), while it is the frequency of occurrence of the regimes that changes in response to the forcing, with some regimes becoming more populated (Palmer, 1999). By analogy, a similar response to forcing has been hypothesized for the WRs in complex GCMs (Palmer, 1999; Corti et al., 1999). Here we use this framework to evaluate the change in the frequency of occurrence of the WRs in the future scenarios, as simulated by the climate models participating in CMIP5 and CMIP6.

The paper is structured as follows: Section 2 describes the Data and Methods used for the analysis; Section 3 shows the results regarding the observed WRs, the model performance and the future projections for the EAT and PAC sectors; in the Discussion

(Section 4) the results are commented with respect to changes in the climate mean state and analysing the connection between the multi-model spread and multiple drivers of the circulation changes; the conclusions are summarized in Section 5.

## 2    Data and methods

### 2.1    Data

An ensemble of GCMs simulations being part of the Coupled Model Intercomparison Project - Phase 5 (CMIP5 Taylor et al., 2012) and Phase 6 (CMIP6 Eyring et al., 2016) are here analysed. For CMIP6, we considered both the historical and four future scenarios simulations with different levels of anthropogenic carbon dioxide emissions along the 21st century: SSP1-2.6, SSP2-4.5, SSP3-7.0, SSP5-8.5 (O'Neill et al., 2016). The SSP1-2.6 scenario corresponds to significantly reduced fossil fuel burning by mid-century and a global warming contained at about 2° C, while the SSP5-8.5 is the business-as-usual scenario. SSP2-4.5 and SSP3-7.0 are intermediate scenarios (Meinshausen et al., 2019). For CMIP5, we consider the historical and the most extreme future scenario RCP8.5 (Moss et al., 2010), which features a smaller $CO_2$ concentration by 2100 than SSP5-8.5 but larger than SSP3-7.0 (Meinshausen et al., 2019). We included 33 CMIP6 and 27 CMIP5 models in the analysis of the model performance for the historical simulations (Section 3.2). The results regarding the future scenarios (Section 3.3) are restricted to the models that were available both in the historical simulation and in all future scenarios considered, resulting in 19 models for both CMIP5 and CMIP6. The models and ensemble members used in the analysis are listed in Table 1.

The reference period for historical simulations spans from 1964 to 2014 for CMIP6 (1964 to 2005 for CMIP5). However, the common period 1964-2005 is considered when comparing the performance of the two ensembles (Section 3.2). The reanalysis data from a combination of the European Centre for Medium-Range Weather Forecast (ECMWF) ERA-40 (1964-1978 Uppala et al., 2005) and ERA-Interim (1979-2014 Dee et al., 2011) is used as a reference. Selecting a different reanalysis product as NCEP does not affect the results, as discussed in Fabiano et al. (2020). For the computation of the weather regimes we consider the wintertime (NDJFM) daily mean geopotential height at 500hPa ("data" in the following). For practical reasons, data are first interpolated to a 2.5° x 2.5° grid using a bilinear interpolation. As discussed in Fabiano et al. (2020), since we are interested in the large scale patterns, this does not impact on results. Since in Section 4.2 we evaluate the role of some drivers in determining the inter-model spread of the response in SSP5-8.5 and RCP8.5 scenarios, monthly averages of the atmospheric temperature (ta) at different vertical levels and wind (ua) in the stratosphere are also used.

### 2.2    Methods

#### 2.2.1    Trend and seasonal cycle removal

The geopotential height field shows a clear increasing trend both in the historical and scenario simulations, due to global warming (see Figure S11). Before computing the weather regimes, data are detrended by applying two different methodologies. Data from historical simulations and reanalysis are detrended by removing the linear trend of the area-weighted season-averaged Northern Hemisphere (30-90N) geopotential height time series. We chose to calculate the trend on the Northern Hemisphere

**Table 1.** Models and ensemble members used in the analysis. Hist stands for historical simulation and ssps means that SSP1-2.6, SSP2-4.5, SSP3-7.0, SSP5-8.5 were all analysed for that model/member (for CMIP6).

| CMIP5 | | | CMIP6 | | |
|---|---|---|---|---|---|
| Model | Member | Experiment | Model | Member | Experiment |
| ACCESS1-0 | r1i1p1 | hist, RCP8.5 | ACCESS-CM2 | r1i1p1f1 | hist, ssps |
| ACCESS1-3 | r1i1p1 | hist, RCP8.5 | AWI-ESM-1-1-LR | r1i1p1f1 | hist |
| BNU-ESM | r1i1p1 | hist, RCP8.5 | BCC-CSM2-MR | r1i1p1f1 | hist, ssps |
| CMCC-CESM | r1i1p1 | hist, RCP8.5 | BCC-ESM1 | r1i1p1f1 | hist |
| CMCC-CM | r1i1p1 | hist, RCP8.5 | CanESM5 | r1i1p1f1 | hist, ssps |
| CMCC-CMS | r1i1p1 | hist, RCP8.5 | CESM2-FV2 | r1i1p1f1 | hist |
| CNRM-CM5 | r1i1p1 | hist, RCP8.5 | CESM2 | r1i1p1f1 | hist |
| CanESM2 | r1i1p1 | hist, RCP8.5 | CESM2-WACCM-FV2 | r1i1p1f1 | hist |
| FGOALS-g2 | r1i1p1 | hist, RCP8.5 | CESM2-WACCM | r1i1p1f1 | hist, ssps |
| GFDL-CM3 | r1i1p1 | hist, RCP8.5 | CNRM-CM6-1-HR | r1i1p1f2 | hist, ssps |
| GFDL-ESM2G | r1i1p1 | hist | CNRM-CM6-1 | r1i1p1f2 | hist, ssps |
| HadGEM2-CC | r1i1p1 | hist, RCP8.5 | CNRM-ESM2-1 | r1i1p1f2 | hist, ssps |
| HadGEM2-ES | r1i1p1 | hist | EC-Earth3 | r1i1p1f1 | hist, ssps |
| IPSL-CM5A-LR | r1i1p1 | hist | FGOALS-f3-L | r1i1p1f1 | hist |
| IPSL-CM5A-MR | r1i1p1 | hist | FGOALS-g3 | r1i1p1f1 | hist, ssps |
| IPSL-CM5B-LR | r1i1p1 | hist | GFDL-CM4 | r1i1p1f1 | hist |
| MIROC-ESM-CHEM | r1i1p1 | hist, RCP8.5 | GISS-E2-1-G | r1i1p1f1 | hist |
| MIROC-ESM | r1i1p1 | hist, RCP8.5 | GISS-E2-1-G | r1i1p1f2 | hist |
| MIROC5 | r1i1p1 | hist, RCP8.5 | HadGEM3-GC31-LL | r1i1p1f3 | hist |
| MPI-ESM-LR | r1i1p1 | hist | HadGEM3-GC31-MM | r1i1p1f3 | hist |
| MPI-ESM-MR | r1i1p1 | hist, RCP8.5 | INM-CM4-8 | r1i1p1f1 | hist, ssps |
| MPI-ESM-P | r1i1p1 | hist | INM-CM5-0 | r1i1p1f1 | hist, ssps |
| MRI-CGCM3 | r1i1p1 | hist, RCP8.5 | IPSL-CM6A-LR | r1i1p1f1 | hist, ssps |
| MRI-ESM1 | r1i1p1 | hist, RCP8.5 | KACE-1-0-G | r1i1p1f1 | hist |
| NorESM1-M | r1i1p1 | hist, RCP8.5 | MIROC6 | r1i1p1f1 | hist, ssps |
| bcc-csm1-1-m | r1i1p1 | hist | MPI-ESM-1-2-HAM | r1i1p1f1 | hist |
| bcc-csm1-1 | r1i1p1 | hist, RCP8.5 | MPI-ESM1-2-HR | r1i1p1f1 | hist, ssps |
| | | | MPI-ESM1-2-LR | r1i1p1f1 | hist, ssps |
| | | | MRI-ESM2-0 | r1i1p1f1 | hist, ssps |
| | | | NorESM2-LM | r1i1p1f1 | hist, ssps |
| | | | NorESM2-MM | r1i1p1f1 | hist, ssps |
| | | | TaiESM1 | r1i1p1f1 | hist |
| | | | UKESM1-0-LL | r1i1p1f2 | hist, ssps |

(30N-90N) - and not on the separate EAT and PAC domains - in order to retain possible decadal basin-wide fluctuations. Anyway, the difference in the future trends when considering the whole hemispheric or the sectorial averages is very small (see right panel in Figure S11). For future scenarios, the detrending is implemented by fitting a third-order polynomial to the above mentioned Northern Hemisphere time series and removing this from the geopotential height field: this is done to suitably fit the acceleration in the global increase in geopotential height seen in the second half of the century. Once the trends are removed, the mean seasonal cycle is subtracted from the data to obtain detrended daily geopotential height anomalies ("anomalies" in the following). The seasonal cycle is computed averaging the data day-by-day at each grid point and applying a 20-day running mean to remove higher frequency fluctuations. It is worth noting that the above-defined average seasonal cycle computed in the historical simulations might differ from the seasonal cycle found in scenarios. Since these differences are part of the change in the midlatitude circulation, it is important to take them into account. Therefore, for each model, the mean seasonal cycle is computed in the reference period of the historical simulation (1964-2014 for CMIP6, 1964-2005 for CMIP5).

### 2.2.2 Weather regimes computation

The weather regimes are computed using the WRtool Python package (Fabiano et al., 2020). We focus here on latitudes between 30° and 90° N and consider separately two longitudinal sectors: the Euro-Atlantic (EAT, from 80°W to 40°E) and the Pacific-North American (PAC, from 140°E to 80°W). The procedure is identical for the two sectors. To reduce dimensionality, an Empirical Orthogonal Function (EOF) decomposition is applied to the observed anomalies, retaining the 4 leading EOFs, which explain 53% and 48% of the total variance for the EAT and PAC sectors respectively. Sensitivity tests performed in Fabiano et al. (2020) for the EAT sector show that the changes in the regime patterns when considering for example 10 EOFs instead of 4 are negligible. The phase space spanned by these EOFs (hereafter "reference phase space") is then used for both the reanalysis and all GCMs simulations: all anomalies are projected onto this reference phase space, obtaining the 4 leading Principal Components (PCs) for the reanalysis dataset and 4 pseudo-PCs for each model simulation. The weather regimes for the reanalysis are computed by applying a K-means clustering algorithm to the PCs. For the EAT sector, we set the number of regimes to 4, as is widely documented in literature (Michelangeli et al., 1995; Cassou, 2008; Dawson et al., 2012; Madonna et al., 2017; Strommen et al., 2019; Fabiano et al., 2020). For the PAC sector, we choose 4 clusters as in Straus et al. (2007) and Weisheimer et al. (2014), although a different number of clusters could be a viable alternative as argued in Straus et al. (2007) and favoured by other studies (e.g. Kimoto and Ghil, 1993; Michelangeli et al., 1995; Robertson and Ghil, 1999). Each day is assigned to one of the regimes and we obtain a set of 4 cluster centroids, defined as the average of all PCs assigned to a certain cluster. The cluster centroids obtained for the reanalysis are referred to as "reference centroids" in the following. The regime pattern is defined as the composite of all anomalies assigned to a certain regime.

For the models, we follow two different approaches to assign each day to a specific regime and, accordingly, two regime types are defined:

– *Computed regimes*: the K-means clustering is performed on the pseudo-PCs and 4 simulated cluster centroids are ob-
tained, as in Fabiano et al. (2020). These *computed regimes* are calculated for the historical simulations only, in order to
compare observed and simulated regimes structure and to assess possible model deficiencies.

    – *Projected regimes*: the K-means algorithm is not applied, but each anomaly in the reference phase space is attributed
to the closest reanalysis reference centroid. In this way, the regimes are consistently defined for all simulations and the
variability in the clustering itself as a possible source of noise is ruled out. The *projected regimes* are used to compare
the regime frequencies and persistence across different simulations/scenarios within a common reference framework.

### 2.2.3 Metrics

Here below a set of metrics used in Section 3 is defined:

    – *Taylor diagram.* A Taylor diagram (Taylor, 2001) is used as a synthetic metric to evaluate how the simulated regime
patterns resemble the observed ones. The Taylor diagram consists of a polar plot showing the spatial correlation between
the simulated and observed patterns (angular axis) and their standard deviation (radial axis, in units of the observed
standard deviation). Due to the geometrical construction, the linear distance between the simulation and the observation
is the centered-pattern RMS (with bias subtracted).

    – *Variance ratio.* The variance ratio is defined as the ratio of the average inter-cluster squared distance to the mean intra-
cluster variance. In cluster analysis, a larger value of this ratio is generally desirable, indicating that the clusters are well
separated from each other. For WRs, the distance from the observed variance ratio is an indicator of the overall model
performance in simulating the regime dynamics (Fabiano et al., 2020).

    – *Regime frequency.* The regime frequency over a certain period is defined as the fraction of days assigned to a certain
regime in that period. Accordingly, the seasonal regime frequency is a time series indicating the fraction of days assigned
to a certain regime in each season. In order to estimate each model performance, the "absolute frequency bias" is defined
as the absolute difference between the simulated and observed regime frequency, averaged over all regimes.

    – *Regime persistence.* The regime persistence is the average duration in days of a given regime event. A regime event is a
set of consecutive days assigned to the same regime. We relaxed this definition to allow for single day departures from
the regime state, thus a regime event is ended only when two consecutive days are assigned to different regimes.

The Taylor diagram, the variance ratio and the bias in regime frequency – calculated for the computed regimes of the
historical simulations – are used to evaluate the ability of climate models in reproducing weather regimes (Sections 3.2). The
change of the projected regimes' frequencies and persistence in future scenarios will be analysed in Section 3.3.

    The observed interannual variability of the regime frequencies and persistence has been estimated to be about 11% and
2.5 days respectively, averaged over all regimes. Therefore, in order to assess significant long-term changes we average these
185 quantities over the following periods: 1964-2014 for the CMIP6 historical runs (1964-2005 for CMIP5) and 2050-2100 for

the scenarios. The variability on a 50-yr window has been estimated as the standard deviation of the mean ($\sigma/\sqrt{n-1}$) of the seasonal frequency and persistence as 1.6% and 0.3 days respectively. However, the actual variability on these scales might be larger than this due to decadal basin-wide fluctuations. For the frequencies, in addition to the differences between the scenario and historical simulations in specific time-windows, the trends in the 2015-2100 period of the scenarios are computed. In order to do this, a 10-year running mean to the time series is applied.

## 3 Results

### 3.1 Observed regimes

The regime centroids obtained from the reanalysis (NDJFM, 1964-2014) for the Euro-Atlantic sector and the Pacific-North America are shown in Figure 1. For the EAT sector, the four regimes are the two phases of the North Atlantic Oscillation (NAO+, NAO-), the Scandinavian Blocking (SBL) and the Atlantic Ridge (AR). The patterns are consistent with those obtained considering different periods and using a different definition for boreal winter (i.e. DJF or DJFM) (Dawson et al., 2012; Fabiano et al., 2020; Cassou, 2008). A close correspondence exists among these regimes, the structure of the North Atlantic jet stream and climatic conditions over Europe. The positive (negative) NAO is related to a central (southern) jet position, whereas a northward displacement of the jet is linked to the Atlantic Ridge regime (Madonna et al., 2017; Fabiano et al., 2020). The SBL regime is related to a high pressure over Scandinavia and corresponds to a tilted jet structure from SW to NE.

The four regimes in the Pacific sector are: the Pacific Trough (PT) (Straus et al., 2007) - the Rockies Ridge in Casola and Wallace (2007) -, the positive and negative phases of the Pacific-North American patterns (PNA+, PNA-; e.g. Wallace and Gutzler, 1981; Barnston and Livezey, 1987), and the Bering Ridge (BR) - also known as Alaskan Ridge - characterised by a blocked flow (Renwick and Wallace, 1996; Smyth et al., 1999; Straus et al., 2007; Casola and Wallace, 2007). The observed regime patterns over the Pacific are consistent with those found by Casola and Wallace (2007) and Weisheimer et al. (2014), although these authors considered different periods and data sets. All four WRs over the PAC sector can be seen as different phases of a Rossby wave train extending from the Pacific towards the North American continent. The variability of such quasi-stationary patterns is modulated by both the interaction of the midlatitude jet with the orography and the forcing by the convection over the equatorial and tropical Pacific, which acts as a Rossby wave source (Trenberth 1998). The PNA+ pattern is in fact in phase with the barotropic response obtained from the interaction of the westerly jet with the orographic forcing provided by the Rocky mountains, and therefore it is associated with an enhanced ridge-through pattern over North America. The PT regime, which is strongly correlated with positive ENSO (Straus et al., 2007; Casola and Wallace, 2007; Weisheimer et al., 2014), exhibits an eastward shift compared to the PNA+. This eastward shift is related to the upper-tropospheric divergence caused by the enhanced convection over the Pacific during the positive ENSO events, which acts as an additional thermal Rossby wave source (Straus and Shukla, 2002). The PNA- and BR patterns appear as out of phase with the PNA+ and PT respectively and have been found to be correlated with La Niña events (Straus et al., 2007; Weisheimer et al., 2014).

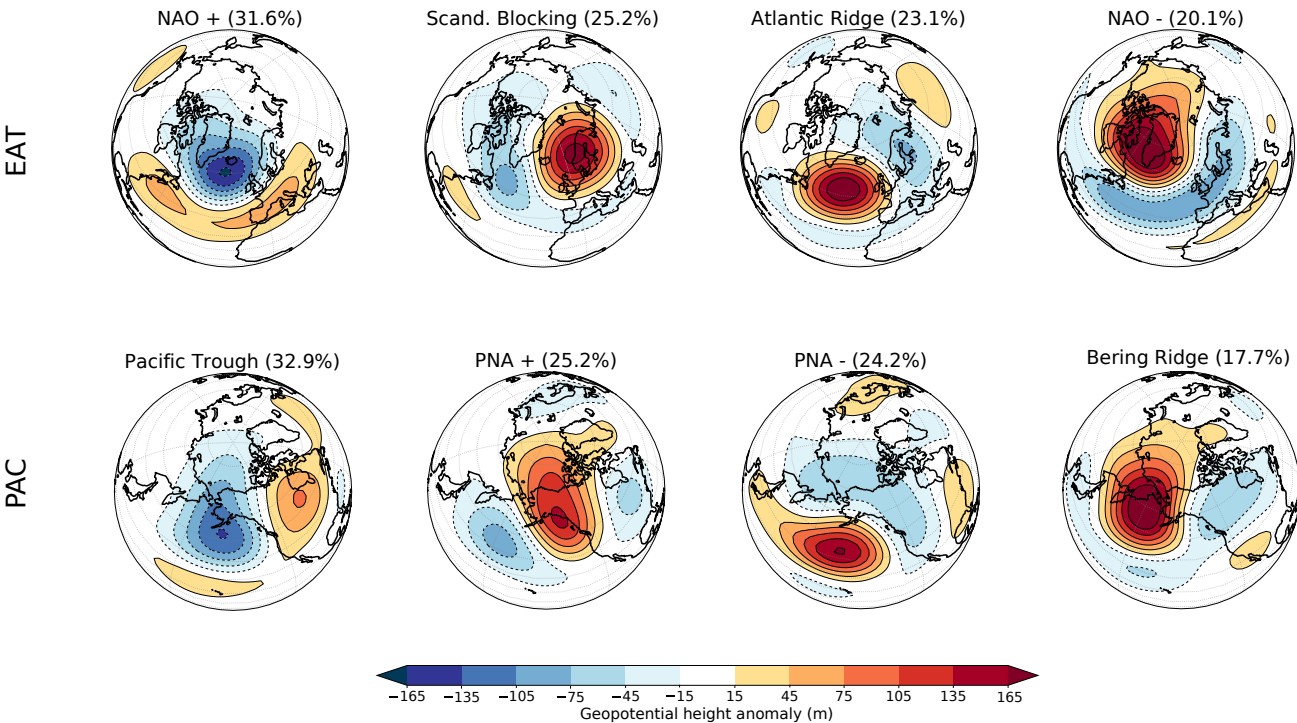

**Figure 1.** Regime patterns for the Euro-Atlantic (top row) and Pacific-North American regimes (bottom row), obtained from the reanalysis (1964-2014, NDJFM). The observed regime frequencies are indicated in the subplot titles. Note that the projection is centered in the Atlantic (Pacific) for EAT (PAC) regimes.

## 3.2 Models' performance

In this subsection the model performance in reproducing the observed weather regimes is assessed. The computed regimes of the historical simulations (for CMIP5 and CMIP6 models) are considered, and the models' performance is evaluated in
terms of the regime centroids, regime frequency bias and variance ratio. The results are shown in Figures 2 and 3. Also, a complementary indication of the relative performance of CMIP5 and CMIP6 models is given in Table 2, which shows the number of models developed by the same institution that improve from CMIP5 to CMIP6 for the three metrics.

Figure 2 displays a set of Taylor diagrams in which the simulated regimes are compared with the observed ones: each simulation is shown by a dot and the overall performance of each ensemble is indicated by the shaded ellipses centered at the
ensemble average and with semi-axes equal to the ensemble standard error. For the EAT sector (Figure 2, first row), the CMIP6 ensemble shows an improvement with respect to the CMIP5 counterpart for all regimes, although the intermodel spread is quite large and the ellipses significantly overlap (apart for NAO-).

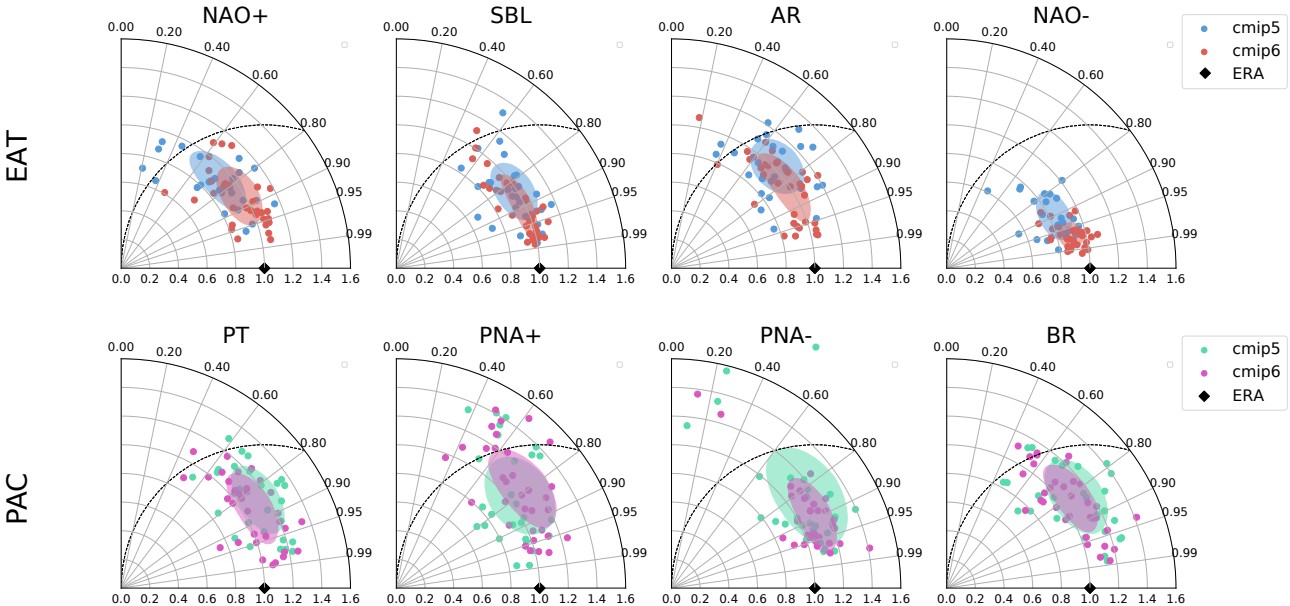

**Figure 2.** Taylor diagrams showing the agreement between simulated and observed regime patterns for CMIP5 (blue and green) and CMIP6 (red and pink) models. The shaded ellipses are used to indicate the overall ensemble performance: they are centered at the ensemble mean and have semi-axes equal to the ensemble standard error The simulated patterns are those obtained from the computed regimes of the historical simulations in the common period 1964-2005. The observed patterns are those computed from the reanalysis.

For the PAC regimes (Figure 2, second row), the difference in the overall performance of the two ensembles is less evident. General improvements are seen in CMIP6 for the negative PNA regime, and, in terms of standard error, for the PT and BR regimes. The performance in simulating the positive PNA regime is comparable in the two ensembles, with a slight worsening in CMIP6 in terms of standard error. PAC regimes appear to be more difficult to capture than EAT ones. This may reflect a larger natural variability in the observed regime structure, as suggested by the smaller variance ratio of the reanalysis for PAC with respect to EAT. Also, the PAC regime patterns might be influenced by the specific history of each model simulation in terms of amplitude and frequency of ENSO events.

The model performance in reproducing the observed variance ratio and regime frequency is shown in Figure 3. For CMIP6 (CMIP5), the average absolute frequency bias is about 2.2% (2.5%) over EAT and 2.3% (2.8%) over PAC, while the variance ratio is around 0.74 (0.72) for EAT and 0.76 (0.79) for PAC. For the variance ratio (left panel), CMIP6 models perform better than CMIP5 ones, with the box getting closer to the observed value (black star) for both EAT and PAC regimes. It is worth noting that – opposite to the EAT sector – models tend to produce larger variance ratios for the PAC regimes than it is observed, which might be due to an excess in the tropically-induced modulation of the PAC regimes in models. An improvement in CMIP6 is also seen for the frequency bias (right panel), more evident in the PAC sector, but also detectable in the reduction of

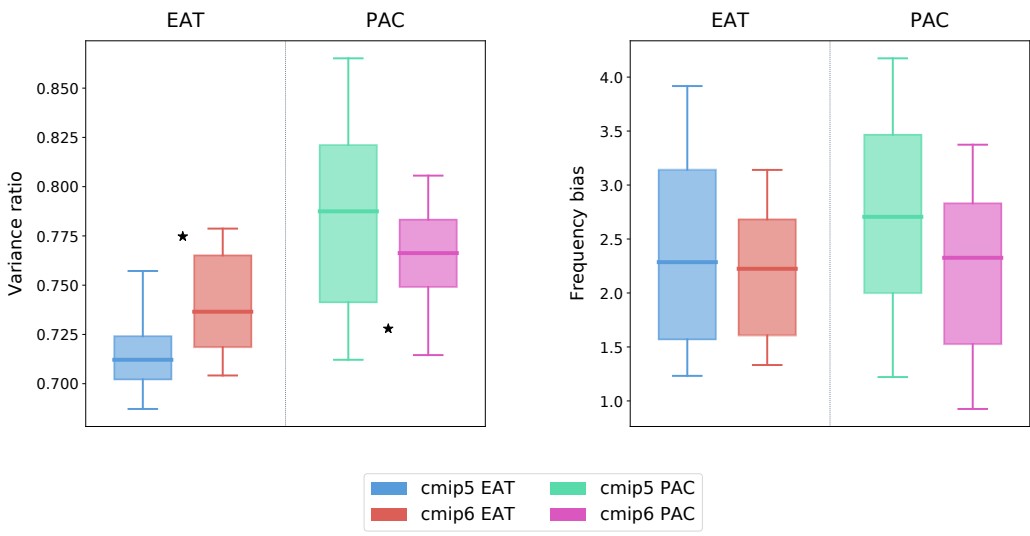

**Figure 3.** Variance ratio (left panel) and absolute frequency bias (right panel) for EAT and PAC regimes. CMIP5 models are indicated by blue (green for PAC) dots and CMIP6 by red (pink for PAC) boxes. The observed values of the variance ratio in the two sectors are indicated by the black stars.

**Table 2.** Number of models developed by the same institution that improve from CMIP5 to CMIP6 for three metrics: average pattern correlation, average frequency bias and variance ratio. There is a total of 11 institution for which at least one model is available for both CMIP phases. If more model versions are available for a single institution, the average metric among all models is used.

| Metric | EAT | PAC |
|---|---|---|
| Pattern | 9/11 | 6/11 |
| Freq. bias | 6/11 | 9/11 |
| Var. ratio | 10/11 | 9/11 |

the spread of the EAT sector. Given the strong link between WRs and blocking events (Madonna et al., 2017; Fabiano et al., 2020), the reduction of the WR frequency bias is in line with the smaller biases in the blocking frequency observed for CMIP6 models (Davini and D'Andrea, 2020).

The results shown in Figures 2 and 3 are confirmed when looking at the performance of models developed by the same institution in the two phases, reported in Table 2: most models improve from CMIP5 to CMIP6 for all three metrics.

### 3.3 Future scenarios

#### 3.3.1 Regime frequency

We analyse here the changes in the regime frequencies in the future scenarios. As discussed in Section 2.2.2, the frequency of
the projected regimes is considered here, i.e. those computed by attributing models data to the closest reference centroid. By
doing so we avoid that changes in frequency are (even partially) produced by a change in the pattern. Also, since the model
biases in reproducing the observed regime frequencies and patterns are significantly smaller for projected regimes than for
computed regimes (see Supplementary materials, Figures S2 and S3), this choice might lead to a somehow higher confidence
in terms of future projections.

The difference between the regime frequencies for the future and the historical reference periods is shown in the top panels
of Figure 4. Each panel shows whiskers plots representing the multi-model distribution of the regime frequencies for the
reference periods (1964-2014 for CMIP6 historical, 1964-2004 for CMIP5 historical, and 2050-2100 for the scenarios). The
CMIP5 historical frequency is shown in Figure S4 and the linear trends of the regime frequencies for the 2015-2100 period of
the scenarios are shown in Figure S6 in the Supplementary Materials. Figure 5 shows the ensemble mean of the WRs seasonal
frequency anomalies for the CMIP6 scenarios and for the CMIP5 RCP8.5, with respect to the historical regime frequency.

The results show a net increase in the frequency of the NAO+ regime in the future. For CMIP6 the signal strength increases
with increased greenhouse gases concentration: there is a smaller increase for SSP1-2.6, and a progressively larger (and statis-
tically significant) increase for SSP2-4.5, SSP3-7.0 and SSP5-8.5. For the two most extreme scenarios the signal is robust, with
the first quartiles of the 50-yr reference period ending up above the historical 90th percentile. Consistently with this picture,
the trend continues till the end of the simulations for the extreme scenarios, while SSP1-2.6 and SSP2-4.5 stabilize before the
end of the century (Figure 5, left panel). The net increases in the NAO+ frequency are confirmed by the positive trends over the
whole 2015-2100 scenario simulations (see also Figure S6, Supplementary Material). The behaviour observed in RCP8.5 of
CMIP5 is in general agreement with CMIP6, though the amplitude of the change is largely reduced with respect to SSP5-8.5,
in particular in the last part of the century, and the difference is not statistically significant with respect to the historical period.

The increase in the NAO+ frequency is accompanied by a general decrease in the AR and SBL frequency. For SBL the signal
is robust for SSP3-7.0 and SSP5-8.5, for which the box stands almost completely below the historical median frequency and the
difference with respect to the historical is statistically significant. The two moderate scenarios also show a small decrease in the
frequency, but the signal is much smaller. The AR regime (Fig 4, bottom left panel) is characterized by strong and statistically
significant reduction in the frequency for all scenarios, as shown by the future boxes staying entirely below the historical 10th
percentile. RCP8.5 shows consistent results for SBL, whose seasonal frequency projection fits remarkably well the SSP5-8.5
one, at least in the second half of the century (Figure 5). For AR the RCP8.5 signal is consistent but reduced in amplitude and
with a larger inter-model spread with respect to both SSP3-7.0 and SSP5-8.5.

The NAO- regime is characterized by a more complex response. There is a general tendency for a small increase in the regime
frequency in the future, but unlike the other regimes the signal is slightly stronger in the moderate scenarios. Also, during the
last 20 years the differences between the moderate and extreme scenarios amplify, showing a larger NAO- frequency for SSP1-

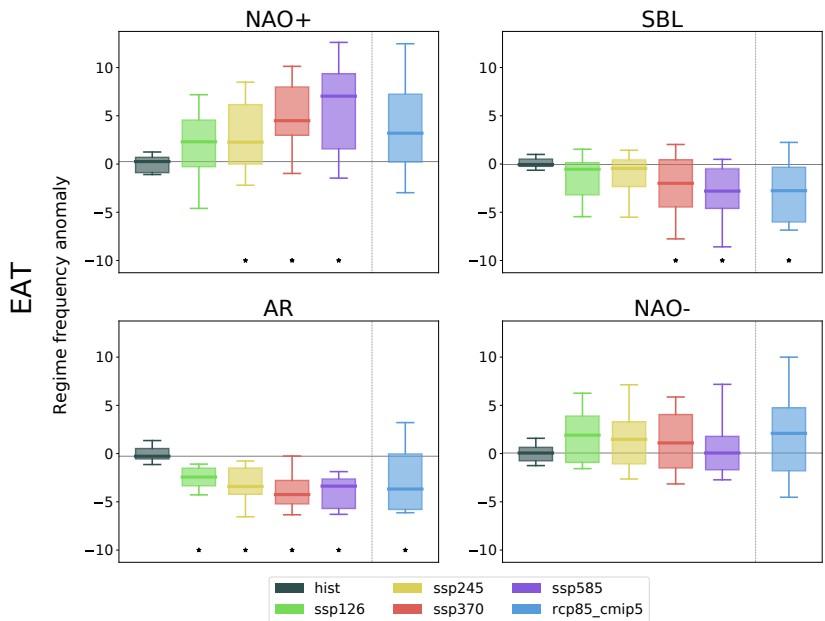

**Figure 4.** Whiskers plot of the multi-model distribution of WRs frequency in the CMIP6 historical (1964-2014) and future (2050-2100) reference periods, with respect to the historical regime frequency. The boxes indicate the first and third quartiles, the horizontal bar indicates the median, and the top and bottom bars indicate the 10 and 90 percentiles. The black star at the bottom indicates a significant difference at the 95% level with respect to the historical distribution using a Welch's t-test.

2.6 and SSP2-4.5 and barely any change for SSP3-7.0 and SSP5-8.5 with respect to the historical period. RCP8.5 shows a consistent trend in the future and a final frequency increase larger than most CMIP6 scenarios, but still not significant with respect to the historical period. Also apparent from Figure 5 are the oscillations in the multi-model mean which appear to be quite in phase up to about 2060. These might be related to the aerosol forcing, which has been hypothesized to have driven the observed AMV oscillations (Zhang et al., 2013; Qin et al., 2020). However, it is not clear whether a similar process might be at work for the future scenario period and further analysis on this topic will be carried out in a different study.

The change in the PAC regime frequencies is shown in Figure 6. The main changes are seen for the Pacific Trough and Bering Ridge regimes. The PT regime shows a net and statistically significant increase in frequency in all future scenarios, with some differences between them but no clear dependence on the future forcing. The RCP8.5 projected PT frequency for the future is slightly larger than the CMIP6 scenarios, but also shows a larger spread. The BR regime is projected to decrease its frequency in the CMIP6 scenarios, with a more robust decrease in the extreme ones. RCP8.5 projections show a larger reduction than the CMIP6 scenarios. The two PNA regimes show smaller and not significant variations in the future frequencies. However, whereas the PNA+ response is consistent with no change at all for all scenarios, the PNA- shows a variation of the response from negative to zero change in frequency ranging from the smallest to the largest greenhouse gases concentrations. The

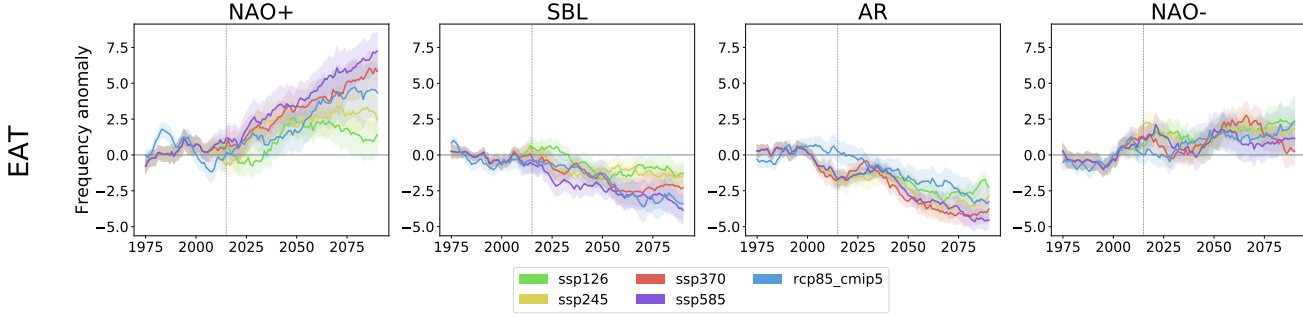

**Figure 5.** Ensemble means of the WRs seasonal frequency anomalies for the CMIP6 scenarios and the CMIP5 RCP8.5, with respect to the historical regime frequency. A 20-yr running mean has been applied to the time series. Shading indicates the standard error of each ensemble. A corresponding figure for the PAC sector is Figure S10 in the Supplementary Materials.

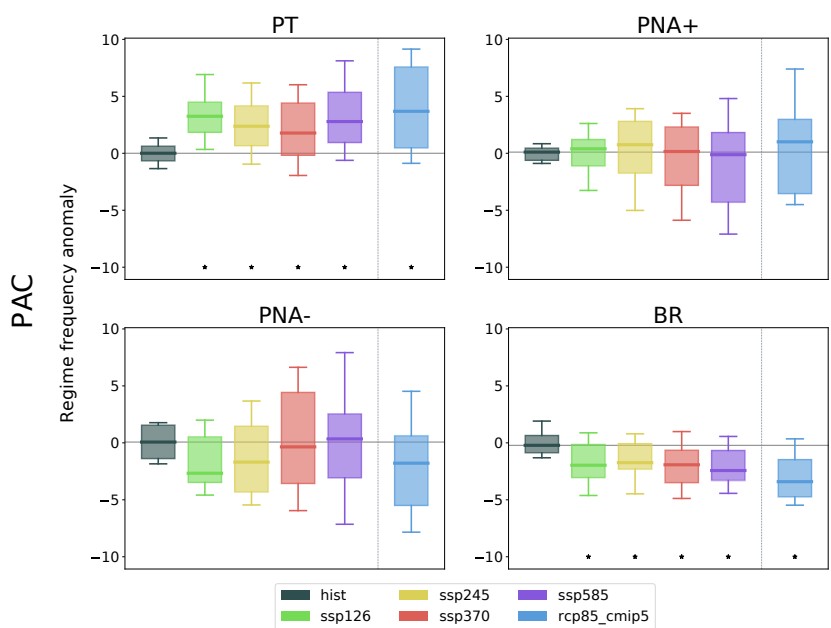

**Figure 6.** As Figure 4, but for PAC regimes.

RCP8.5 ensemble shows a reduction in the future PNA- frequency, in contrast with SSP5-8.5. Interestingly, even if the two PNA regimes frequencies are not changing significantly, the spread in the model response increases proportionally to the $CO_2$ forcing. This suggests that models respond linearly to increased forcing, but that there is a large intermodel spread in the response.

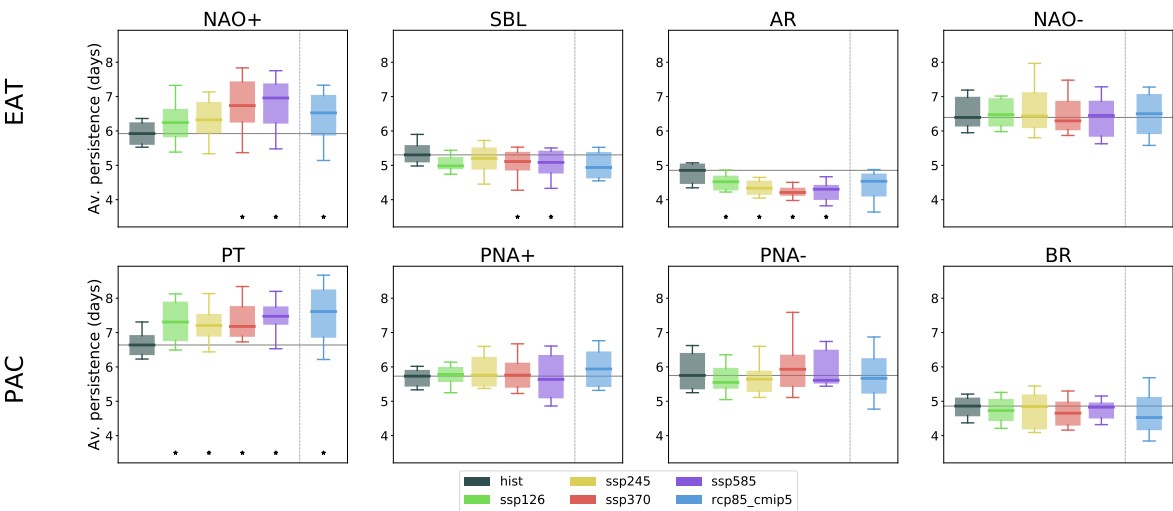

**Figure 7.** Average regime duration in days for the historical runs and future scenarios. Boxes are defined for the same periods as in 4, and the black stars at the bottom indicate significant variation with respect to the historical period.

### 3.3.2 Regime persistence

The average regime persistence also changes in the future according to CMIP6 and CMIP5 models. As shown in Figure 7 (first row), the change in the average regime duration (days) for the EAT sector is generally consistent with the direction of changes in the regime frequency. The NAO+ regime is expected to have a longer duration, with the largest increase in the average number of days per regime event up to about one day in the SSP5-8.5 scenario. In the two extreme scenarios, NAO+ takes the place of NAO- as the regime with longest average duration. Concurrently with the NAO+ increase, we observe a large decrease

in the average duration of the AR regime and a small - but statistically significant - decrease for the SBL regime. No significant change is seen for NAO-. RCP8.5 confirms these tendencies, though the amplitude of the NAO+ and AR change is reduced with respect to CMIP6, and changes for SBL and AR are not statistically significant. In the PAC sector (Figure 7, second row), the PT regime shows a substantial response, with a net and statistically significant increase in the average regime duration in all future scenarios. The SSP5-8.5 scenario shows the largest increase, reaching an average duration of about 7.5 days in the

2050-2100 period. The response to the RCP8.5 scenario is consistent with that to SSP5-8.5, but the model spread is larger. The other regimes do not show a clear variation of the persistence in the future in the CMIP6 ensemble, while RCP8.5 projected the BR persistence to slightly decrease in the future.

Figure 8 shows the number of regime events per 100 days. The changes in the regime frequencies might be seen as the combined effect of the changes in the regime persistence and the changes in the number of regime events. For the EAT sector,

both have a comparable role in the frequency change of AR and SBL, while the increased persistence seems the main factor

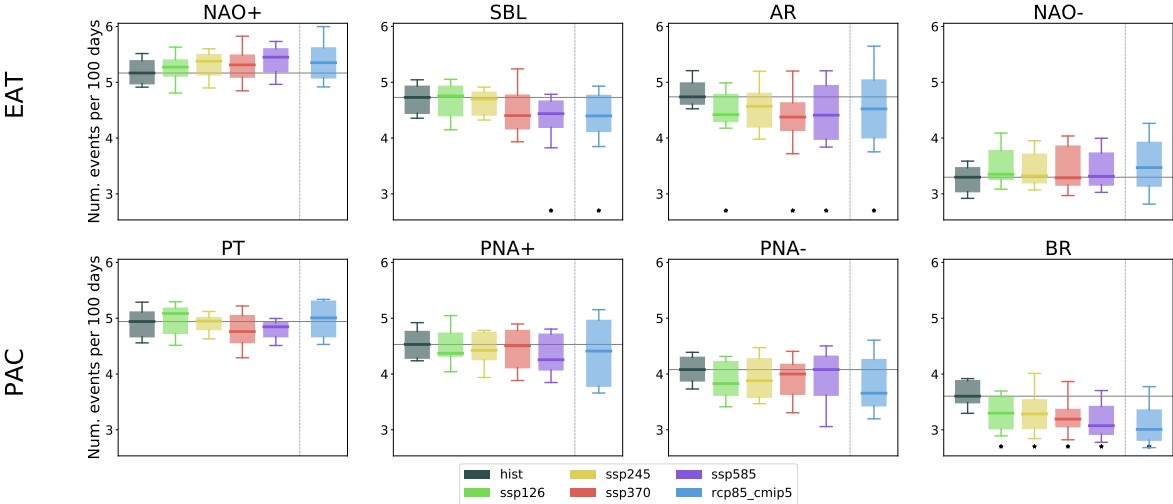

**Figure 8.** Number of regime events per 100 days for the historical runs and future scenarios. Boxes are defined for the same periods as in 4, and the black stars at the bottom indicate significant variation with respect to the historical period.

in the NAO+ change. For the PAC sector, the increase in PT frequency is driven by longer persistence, despite no significant change in the number of events, while the opposite is true for the change in the BR regime frequency.

## 4 Discussion

The projected changes in the regime frequencies in the future scenarios give a clear picture of the evolution of the variability in the mid-latitudes dynamics. For the EAT sector, the results shown in Section 3.3 are consistent with a zonalization of the mid-latitude circulation and a squeezing of the eddy-driven jet distribution around its central position (Peings et al., 2018; Oudar et al., 2020). There is generally a good correspondence between the Euro-Atlantic WRs and the North-Atlantic jet latitude index (Madonna et al., 2017; Fabiano et al., 2020). An increase in the NAO+ frequency corresponds to a more frequent central jet position, while the decrease in SBL means a lower probability of a tilted jet and a reduction in the spread of the distribution. This is also consistent with the reduced variance in terms of latitudinal shifts of the jet, observed by Barnes and Polvani (2013) on the CMIP5 ensemble. Also, a less frequent AR regime would mean a reduction in the northward peak of the jet latitude distribution. The decrease in the AR and SBL frequency is in agreement with the predicted decrease in the blocking frequency over Europe, observed on CMIP5 and CMIP6 models (Davini and D'Andrea, 2020). The change of WR frequency observed for RCP8.5 is consistent with the result by Cattiaux et al. (2013a), although they observe a larger increase of NAO-, which may be due to a different treatment of the climatological mean state.

The strong increase in the NAO+ regime frequency is in line with the change of storm track activity in CMIP6 projections analysed by Harvey et al. (2020), which shows an intensification of the activity over the North-Atlantic and central/northern

Europe, with a center on the British Isles and an increased penetration of perturbations into the continent. A corresponding decrease of perturbations at very high latitudes is also in line with a decrease in the AR regime, which tends to push the jet poleward. In terms of impacts, NAO+ drives mild temperatures over the Eurasian continent and a North-South precipitation dipole with increased precipitation over Northern Europe and dry conditions over South Europe. For negative NAO, colder temperatures are found in Northern Europe, and the precipitation dipole is reversed, with increased precipitation in the South (Yiou and Nogaj, 2004). The increased NAO+ frequency in the future would thus lead to higher winter precipitation in the Northern part of the continent, with a concomitant lower precipitation and higher risk of droughts over the Mediterranean region. At the same time, the increased persistence may increase the risk of flooding in Northern Europe.

With regards to the PAC sector, the increased frequency of the PT regime and the reduction in the BR regime are consistent with the projected decrease in blocking frequency in the Bering Strait region (Davini and D'Andrea, 2020). The PT regime is characterized by a positive geopotential anomaly over central Canada, and its increased frequency in the future may be linked with the projected increase in the waviness index observed by Peings et al. (2017) in this region. Also, a strong decrease of the storm track activity under SSP5-8.5 has been observed over the whole Northern American continent, and an increase in the central North Pacific (Harvey et al., 2020). This agrees well with the prediction of an increased PT regime, that blocks the entrance of perturbations in the continent. In the observations, the PT regime tends to be more frequent during positive ENSO (Straus and Shukla, 2002; Weisheimer et al., 2014), therefore its increase in the future may be linked with an increased Niño-like forcing. A recent study of ENSO occurrence in CMIP6 projections (Fredriksen et al., 2020) shows a tendency for an increase of ENSO variability under global warming, and interestingly this change appears to be mostly related to an increase in positive El Niño events.

### 4.1 Relation with changes in the mean state

The change in the regime frequency is inextricably linked to a modification of the mean geopotential height at 500 hPa. On the one hand, one can explain variations in the frequencies as the result of a global shift of the climate state towards one regime. On the other hand a change in the mean state can be interpreted as the side effect of an increase in the occurrence of some weather regimes (e.g. NAO+ and PT) and a corresponding decrease in the frequency of others (e.g. SB and BR). The change in the mean state of the geopotential height at 500 hPa during the extended boreal winter (NDJFM) for the 2015-2100 period in the SSP5-8.5 scenario multi-model ensemble simulations is analysed by taking deviations from the third-order polynomial fit of the area-weighted season-averaged Northern Hemisphere (30-90 N) time series. The projected change in the geopotential height depends on both latitude and longitude. For further insight, the multi-model mean response is split in two parts:

  – the zonal mean trend anomaly, shown in Figure 9;

  – the local departures from the zonal mean trend (or, equivalently, the trend of the stationary eddies), shown in Figure 10.

The multi-model average of the zonal mean trend anomaly shows a larger increase in the geopotential at high latitudes and a smaller change at mid-latitudes, peaking at about 60 N. Restricting the analysis to the EAT and PAC sectors gives similar results, but with an intensification of the negative anomaly at mid-latitude in EAT and a southward shift of the negative peak

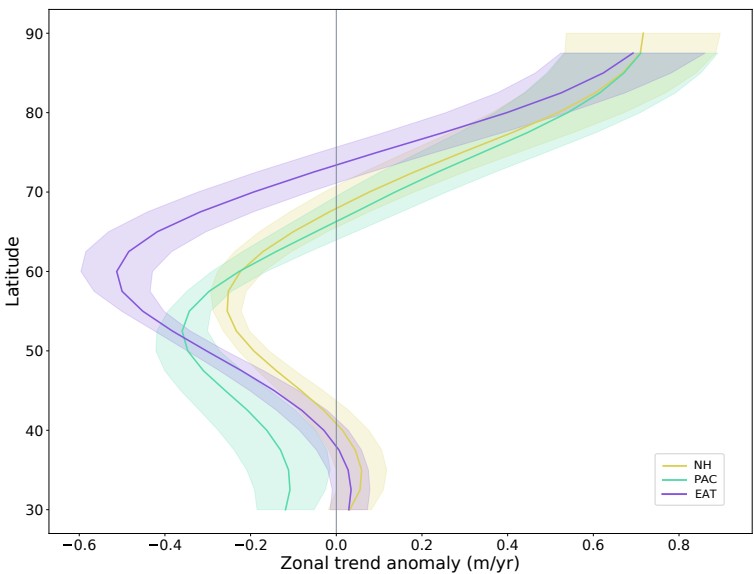

**Figure 9.** Multi-model average of the zonal-mean trend anomaly of the geopotential height at 500 hPa for SSP5-8.5. The zonal trend anomaly is shown for the Northern Hemisphere from 30 to 90N (NH, yellow) and for the two sectors analysed here, PAC (green) and EAT (purple), after removing the global NH trend.

in PAC. The trend of the stationary eddies provides further insight in the mean state change. The negative trend in the North Atlantic, west of the British Isles and south of Iceland, is consistent with a more frequent occurrence of the NAO+ regime and a decrease in the frequency of the AR regime described in Section 3.3. A positive trend over the Mediterranean region and the development of two highs, one over central Northern Canada and the other over the whole Asian continent can be noted as 370 well. This picture is consistent with an increase of the geopotential at high latitudes and the concurrent eastward shift in phase of the stationary waves already observed in CMIP5 models (Wills et al., 2019; Simpson et al., 2014). The polar high is linked to the increased temperatures in the region, due to Arctic amplification, and the shift in phase may be due to the decrease of the dominant zonal wavenumber of stationary waves with global warming, as found by (Wills et al., 2019).

## 4.2 Potential drivers of future circulation changes

Although future changes in regime frequencies and average persistence times are apparent in the second half of the 21st century when multi-model ensemble means are considered, a considerable spread in the model response is evident as well and may be linked to differences in the model climate feedbacks. We here analyse possible drivers of this spread in the SSP5-8.5 and RCP8.5 scenarios. A method to investigate the model spread is to decompose the mid-latitude future response into different

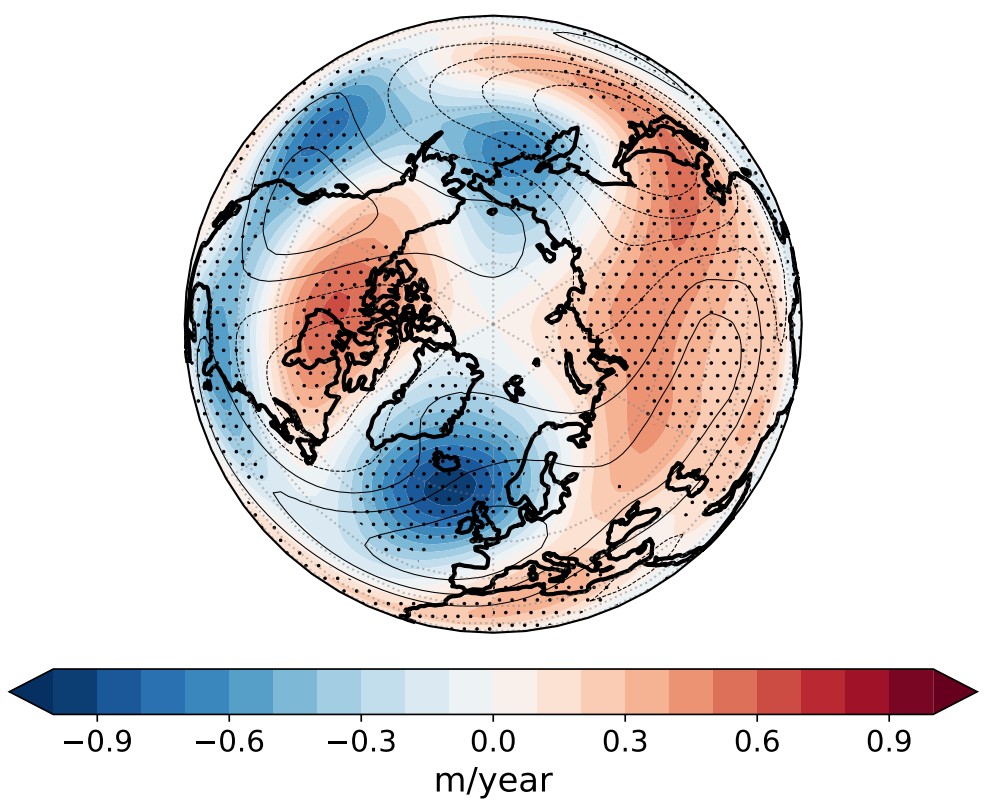

**Figure 10.** Multi-model average of the trends in the stationary eddies (shading) and multi-model mean of the stationary eddies for the historical simulations (contour). Hatching indicates where the 80% of the models agree on the sign of change. The trend in the stationary eddies is equivalent to the residual local trend, after removing the global and zonal components.

components related to the differential warming of the Earth's atmosphere. A set of indices that (in principle) have the potential
to affect the mid-latitude circulation and the WR frequencies are selected:

- *UTW*: the upper tropospheric warming, computed as the temperature trend in the tropical upper troposphere (averaged between 20S and 20N and from 400 to 150 hPa);

- *AA*: the Arctic amplification, computed as the temperature trend in the Arctic lower troposphere (averaged between 60N and 90 N and from 1000 to 700 hPa);

- *PST*: the polar stratospheric temperature, i.e. the temperature trend averaged between 70 and 90 N and from 250 and 30 hPa;

- *TNAW*: the surface warming trend in the tropical North Atlantic (averaged between 80W and 10E, 10N and 30N);

- *SPGW*: the surface warming trend in the North Atlantic subpolar gyre (averaged between 60W and 20W, 40N and 70N).

The first three indices have already been considered as potential drivers of the circulation changes in future scenarios by Zappa and Shepherd (2017), Peings et al. (2017, 2018) and Oudar et al. (2020). The above-defined indices have been computed as trends over 2015-2100 (2006-2100) for SSP5-8.5 (RCP8.5). In order to explore the potential links between them and the projected change in regime frequencies (Section 3.3), a multi-linear regression analysis has been performed. This analysis aims at finding significant relationships between the set of potential drivers and the WR frequency trends. Of course, a significant correlation between two quantities does not demonstrate the existence of a causal link - as they might be responding to a common external forcing - nor does it give indication on the direction of such a link. Nevertheless, it can provide insight into the interconnection between mid-latitude climate variability and large-scale changes in GCMs that deserve further investigation.

Since some of the indices are highly correlated (see Table S3, Supplementary Material), all have been divided by the global surface warming, i.e. the global trend of the atmospheric surface temperature (tas) during 2015-2100. After this operation, moderate correlations remain between AA and TNAW (0.28), TNAW and SPGW (0.28), SPGW and UTW (-0.25), while other correlations have absolute values below 0.2 (see Supplementary Materials, Table S2). All indices have been standardized to zero mean and unit variance before performing the regressions. The frequency trends were also divided by the global warming.

Figure 10 shows the optimal sets of 2 and 3 indices to fit the regime frequency trends in each sector: the columns represent the different regimes and the rows the indices. The optimal set is the combination of the potential drivers listed above with the highest R2 score (i.e. the highest explained variance). The regression coefficients are shown in color code and the respective statistical significance is indicated by the white circles (big circle: p < 0.01; small circle: p < 0.05). However, the effective size of the sample may be smaller than the 38 models considered here (19 for both SSP5-8.5 and RCP8.5), since some models are closely related to each other and this might lower the significance of the regressions. The overall R2 score of the regressions are 0.28 (0.21) and 0.34 (0.30) for the 2 and 3 indices set of the EAT (PAC) sector. However, the score varies strongly among the regimes and it is higher for NAO+ (0.5) and NAO- (0.55) and very low for SBL and AR (0.1-0.2). On PAC the score is more uniform, between 0.25 and 0.4 for all regimes (see Figure S12). The regression model with all 5 indices is shown in Figure S13.

For the EAT regimes, the dominant connections are found with the polar stratospheric temperature and the Arctic amplification. The polar stratospheric temperature explains a considerable fraction of the spread in the EAT response, with warmer temperatures driving a decrease in NAO+ states and an increase in NAO-. Indeed, a warm polar stratosphere is linked to a decrease in the polar vortex strength, and a negative NAO index is more commonly observed in response to a weak polar vortex (Baldwin and Dunkerton, 2001; Ambaum and Hoskins, 2002). This is in line with other works that found a significant relation of the PST with the North Atlantic circulation changes (Manzini et al., 2014; Zappa and Shepherd, 2017; Peings et al., 2017). The Arctic amplification goes in the same direction as PST and is linked with a reduction in the NAO+ and an increase in the NAO- frequency, which is consistent with the expected contribution of AA towards a weakening and equatorward shift of the jet (Barnes and Screen, 2015; Peings et al., 2018; Cohen et al., 2019). The other indices only explain a small portion of

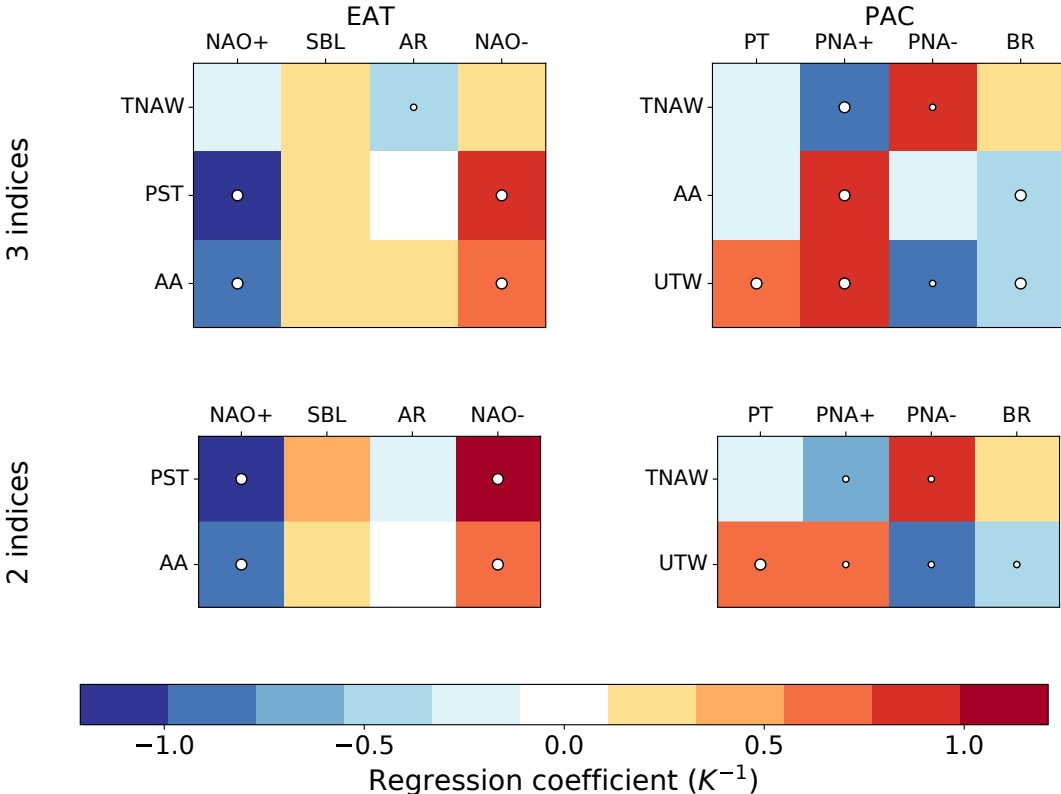

**Figure 11.** Regression coefficients for the best sets of 2 and 3 indices to fit the projected trends of the weather regimes frequencies in the two sectors. The indices have been standardized before performing the multi-linear regression. The regression coefficients are shown in color code and the respective statistical significance is indicated by the white circles (big circle: p < 0.01; small circle: p < 0.05). The standard deviation of the indices before standardization are shown in the Supplementary Material, Table S1.

the remaining variance. The significant negative correlations of TNAW with AR and of SPGW with NAO+ (Figure S13) may indicate the influence of local SSTs on those regimes, although the NAO+ frequency change may also have an impact on the local surface temperature and SPGW. The role of the UTW is less clear, since the only significant positive correlation is with NAO- (Figure S13), but a stronger meridional gradient in the upper troposphere would instead be expected to push towards the

zonalization of the jet (Barnes and Screen, 2015; Peings et al., 2018).

Over the PAC sector, the most significant relations are found with UTW and TNAW, but the 2-indices model explains a smaller fraction of the total variance than for EAT. The UTW is positively correlated with PT and PNA+, and negatively with PNA- and BR, which resembles the influence of positive ENSO on the PAC regimes (Straus and Shukla, 2002). The TNAW has a significant regression coefficient with the two PNA regimes, opposite to that of the UTW, and only small non-significant

regressions with PT and BR. This influence of a warmer tropical North Atlantic on the Pacific regimes may be due to its link with negative ENSO. In fact, there are indications that the strength of the Pacific Walker circulation may be controlled by the

Atlantic ocean temperature (McGregor et al., 2014; Li et al., 2016), with a warmer Atlantic linked to more frequent La Niña-like conditions. However, this significant regression may also show the response to a common external forcing, influencing both the TNAW and the PNA regimes, like a change in the ENSO forcing (Fredriksen et al., 2020). Among the other indices, the AA is linked with an increase in PNA+ and a decrease in BR, while PST and SPGW only have significant positive correlations with PT and BR respectively (Figure S13).

## 5    Conclusions

We proposed here an alternative view of future changes in the atmospheric circulation at Northern mid-latitudes, considering the future trends in weather regimes frequency and persistence over the Euro-Atlantic and Pacific-North American sectors, as projected by CMIP5 and CMIP6 models. The CMIP6 ensemble shows a non negligible improvement in the reproduction of the weather regimes when compared to CMIP5 (Section 3.2). A better simulation of the regime patterns is in particular evident over the EAT sector, however both sectors show an improvement in the other metrics considered (i.e. the variance ratio and the regime frequency bias, see Section 3.2). The model biases in simulating the observed regime centroids, frequencies and variance ratio are known and documented in literature (Dawson et al., 2012; Weisheimer et al., 2014; Strommen et al., 2019; Fabiano et al., 2020). The improvements of CMIP6 models compared to CMIP5 in this respect are therefore encouraging.

Over the EAT sector an increase in the NAO+ frequency and persistence during the second half of the 21st century is observed in all scenarios, with larger changes in the SSP3-7.0 and SSP5-8.5 (Section 3.3.1). This increase is accompanied by a decrease in the AR frequency (and persistence) in all scenarios and a decrease in the SBL frequency, more pronounced in the most extreme scenarios. The NAO- regime shows a small positive trend in all scenarios. These trends are consistent with changes in the mean geopotential height state, that shows an increase at high latitudes and a pronounced eastward shift of the stationary eddies (Section 4.1). A significant fraction of the spread of the model response over the EAT sector is related with the spread in the polar stratospheric temperature and the Arctic amplification in future projections (Section 4.2). The increase of the NAO+ regime is consistent with a squeezing of the jet around the central position (Peings et al., 2018; Oudar et al., 2020) and with a reduced meridional variability of the jet (Barnes and Polvani, 2013).

In the PAC sector the future trends are characterized by an increase in the PT regime occurrence, with a concomitant decrease in the BR regime frequency. The two PNA regimes do not show clear trends in the future scenarios. The intermodel spread in the PAC WR trends correlates significantly with the upper tropospheric warming in the tropics and the warming of the tropical North Atlantic (Section 4.2). The increase in the PT regime frequency indicates a larger relative importance of tropical forcing versus orographic forcing in perturbing the mean flow. The decrease in the BR regime is consistent with changes in the mean state (Section 4.1) and with a decrease in the blocking frequency in the Bering Strait (Davini and D'Andrea, 2020).

The regime perspective presented in this work provides a clear picture of future changes in the wintertime mid-latitude atmospheric circulation and also introduces a suitable framework to study the impact of extreme weather in the future scenarios. The projected change in the regime frequencies are associated with important changes in the temperature and precipitation statistics over different regions. For example an increase in frequency of a strong zonal flow regime (i.e. NAO+) over the

465 Atlantic can lead to an enhanced flood risk due to its connection with stormy weather over northwestern Europe and the British Isles (Yiou and Nogaj, 2004). In this respect it is worth noting that the extreme rainfall in the UK during winter 2013-2014 resulted from this type of atmospheric circulation (Knight et al., 2017) and human-induced climate change was recognised as the major driver of such an extreme (Vautard et al., 2016). Also, the Mediterranean region might suffer from summertime dry spells and heat waves in response to a deficit in precipitation during winter (Vautard et al., 2007), which is more likely to occur

with increased NAO+ frequency.

*Code availability.* The WRtool package is freely available at https://github.com/fedef17/WRtool.

*Author contributions.* FF conducted most of the data analyses and all visualisations and wrote the paper. VM performed part of the data analysis. VM, PD, PG and SC all commented, organized and wrote parts of the paper.

*Competing interests.* The authors declare that they have no conflict of interests.

*Acknowledgements.* The authors acknowledge support by the PRIMAVERA project of the Horizon 2020 Research Programme, funded by the European Commission under Grant Agreement 641727.

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
