# Peer review of "A regime view of future atmospheric circulation changes in Northern mid-latitudes"

_Weather and Climate Dynamics, 2020_

## Referee Comment (RC2) · Anonymous Referee #2 · 23 Oct 2020

This paper deals with the changes in weather regimes in the Euro-Atlantic and Pacific-North American sectors, in both CMIP5 and CMIP6 models. A comparison of simulated and observed weather regimes is done and then the changes in frequency and persistence of the weather regimes are characterized. Potential drivers of those changes are finally discussed. This is a nice paper, well written, with interesting results. The paper is concise, despite the large amount of work necessary to do these analyses. I recommend publication of the paper after some moderate revisions.

Main comments

The weakest part of the paper (but also, potentially, the most interesting one) is probably section 4.2. The analyses themselves are interesting but I think the interpretation of the results should be more cautious. It is not because a significant (?) correlation be-

tween changes in weather regimes frequency and what the authors call the "drivers" is found that a causal relationship exists. The weather regimes could themselves impact these drivers, and other factors, not studied by the authors may physically explain both the changes in weather regimes and in the drivers. More drivers could also have been considered: e.g. many studies have shown the potential impact of SST anomalies (especially in the Tropics, but not only) on large scale circulation, including on weather regime occurrence. I think it is difficult to discuss the potential drivers of weather regime changes without investigating the role of regional SST changes.

I also think that a discussion on the potential causes of the differences seen between CMIP5 and CMIP6, which are sometimes quite large, should be added to the paper. An interesting analysis would be to evaluate whether differences between the changes in weather regimes are linked to differences in the changes of the drivers between CMIP6 and CMIP5 (if the authors are right about the drivers, it should be the case).

Many methodological choices are necessary in a weather regime analysis: e.g. pre-processing through EOF analysis or not, if yes number of principal components retained, precise algorithm (many variants of k-means exist), choice of the number of clusters, whether all the days are classified or not, if not what is the criteria for not classifying some days, which variable is used (slp, zg), how to best take into account the mean increase in zg due to global warming etc. I somewhat understand why the authors don't want to discuss all these aspects, show sensitivity tests etc. The paper would be too long and less interesting. But nevertheless some aspects deserve to be better justified, at least with a sentence. Sensitivity tests could also be added in SI (as the authors decided to add SI to the paper). For example, the authors have chosen to first apply an EOF decomposition to zg and then to retain the 4 leading EOFs. They give no justification for this pre-processing step. The use of principal components for cluster analysis has been common in the past but I suspect that in many old studies, it was more a question of dimension reduction in order to run classification algorithms on computers with limited resources. The use of principal components is also sometimes intended as a kind of filter. If it the case here, the authors should explain their objective(s). It is also unusual to retain such a small number of principal components for clustering, and therefore such a low level of explained variance. I would like to see a discussion on this point and potentially the result of a sensitivity test: e.g. changes in weather regimes without EOF analysis, or with enough principal components to keep at least 90% of the variance etc.

Finally, it would also be interesting to characterize the impact of internal variability on the changes in weather regime occurrence, as signal-to-noise may be small regarding atmospheric circulation changes. It would help to better evaluate the importance of future changes, of the differences between CMIP6 and CMIP5, and between scenarios etc. There are a few CMIP6 models with at least 10 members for which Zg is provided (for future scenarios). It would be very interesting to use the members of these models to characterize the impact of internal variability (for a single scenario, ssp585 for example, this is sufficient).

Specific comments

L15. Please define precisely what is meant by "low-frequency" in the paper.

L103. I think that Table S1 with the models and members used should be in the main document.

L105-108. I'm not sure it is good idea to combine the results of two reanalyses (ERA40 and ERA Interim). Moreover, both of these reanalyses have now been superseded by ERA5, with major improvements in ERA5. I understand that ERA5 is not (yet) available before 1979, but I'm not sure why the authors absolutely need to start in 1964. They could use the period 1979-2015. It would lead to a smaller sample, it is true, but I also think that the hypothesis of a linear trend (section 2.2.1) may be more reasonable on a shorter than on a longer period.

L118. Please explain (in the paper) why you need to "detrend" the data (by the way,

I'm not sure that detrend is the good word considering what is done). And please justify why you use the average on the Northern Hemisphere (30-90N), rather than the averages on the domains of classification, as it is usually done. I think it may have some non-trivial implications, for example if the mean increase in geopotential height, largely controlled by warming, is different between the Euro-Atlantic and Pacific-North American sectors, as it may favor artificially some weather regimes in both domains.

L132-134. See my main comments. Why do the authors only use 4 EOFs for around 50% of explained variance?

L136. I'm not sure that it is a very good justification, but OK...

L142. How is the centroid exactly defined: average, median, real day closest to the average etc?

L145. Are all days classified and why? In some studies, the transition days are not classified, for example.

L146. Computed regimes: how are the computed regimes associated with the observed regimes? Is the associated observed regime the one with the stronger spatial correlation? Are there ambiguities? (e.g. a computed regime that looks like something intermediary between two observed regimes)

L176-177. Not clear to me.

L223-22. I'm not sure to understand the reasoning.

L263. "Further analysis..." I understand, but it is unfortunate. It is a very interesting (and strange) results. Discussing some hypotheses would have been nice. Note that it may be relevant regarding the discussion about the "drivers", as the "drivers" could be implicated in these variations (if they really are "drivers").

L315. I think the mean changes in geopotential height at 500 hPa (maps) should be shown somewhere, maybe as the first figure of the paper.

L320 and legend of figure 8. It is not clear what the "trend" is here. In section 2.2.1 the trend is defined as filtered area-weighted average Northern Hemisphere geopotential height, and used to detrend the local geopotential height. Therefore, if my understanding is correct, here the authors look at the zonal mean "trend of detrended" data, right? It is not clear, and bit awkward from a vocabulary point of view. Please indicate the period used to compute the trends in the legend.

L323: zonal mean trend or zonal mean trend anomaly, as said in the legend?

L351. For the predictors I suppose that you also use the trends, as for the predictands (it is not said explicitly)? What is the period on which the trends are calculated for the predictors and the predictands? 2015-2100 I guess, but it is not explicitly said I think (also in the legend of Fig 10).

Section 4.2. Are there correlations between changes in EAT and PNA regime frequencies? I think knowing that might be useful for the discussion in this section.

L352. Why 2 or 3? Please justify. Are (i) all the models with the ssp585 scenarios used, or, at it seems to be the case based on Table S1 (ii) only the models with the 4 scenarios are used even for this analysis? I think (i) would be much better as it would lead to a larger sample of models, which is quite small with (ii). It is even truer since some of the models are nearly duplicates: models at different resolutions, ESM and AOGCM from the same group etc., which decreases the "effective" sample size.

L362. I don't see figure S7. Are these correlations significant (with the issue of effective sample size mentioned above it is difficult, or impossible, to do the test right, but it is still an interesting indication).

L387. Any idea of the reason(s) that might explain the improvements in CMIP6? Please discuss.
* * *

---

## Author Response (AR1)

We thank the reviewer for the comments and for the deep analysis of the manuscript, that helped to improve the overall quality of our work.

General comments

This study investigates future changes in Atlantic-European and Pacific-North American weather regime occurrence according to CMIP projections. In a first part, the authors evaluate how well the models represent weather regimes (WRs) in their historical simulations compared to reanalysis, with a particular focus on how models from the 6th CMIP phase have improved compared to the 5th phase. In a second part, they investigate how WR frequency and persistence changes by the end of the century. In a last part, they trace these changes back to changes in the atmospheric mean state.

The study provides an important basis for understanding most recent projections of future changes in surface weather from a large-scale dynamics point of view and opens interesting questions for further research. Moreover, the detailed analysis of model biases can be a good guidance for the CMIP community to further improve their models. The paper has a clear and logical structure and is comprehensibly written. The methodological procedure is thorough and transparent. Aside from one major concern, I only have a (relatively large) set of minor comments that should be addressable relatively easily by providing some further explanation or making small adjustments in the text or figures. Therefore, I suggest the paper to be published after considering this major comment and the list of minor comments.

Major comment
• Section 4.2 (Drivers of future circulations): Although I appreciate the attempt to understand the origins of the projected changes in weather regimes (WRs) in more detail, I find this particular analysis in its current form not convincing enough from a causality perspective. In my opinion, this starts with calling the four indices (UTW, AA, PST, NAW) "remote drivers" and using them as statistical predictors for "predicting" WR changes based on linear regression. The reason is that there are strong dynamical links between mid-latitude storm track activity (and thus the WRs) and these indices. For instance, Ambaum and Hoskins, 2002 (http://shorturl.at/ipDO9) suggested a strong coupling between the NAO and the stratospheric polar vortex (which can be used as a proxy for your PST to first order) in the sense that a positive NAO can trigger a strong polar vortex, which in turn can strongly couple with the troposphere and induce persistent periods of positive NAO. Similarly, the effect of weak stratospheric polar vortex states on the troposphere (and thus on WRs) has shown to be strongly influenced by synoptic activity or WR occurrence beforehand (e.g., Kodera et al., 2016, https://doi.org/10.1002/2015JD023359; Domeisen et al., 2020, https://doi.org/10.5194/wcd-2019-16). Along the same lines, Garfinkel et al., 2015 (https://doi.org/10.1002/2015JD023284) showed that for instance SST anomalies (and thus the tropospheric state / storm track activity) can contribute to Arctic lower-stratospheric temperature changes (i.e., your PST). Likewise, I find it surprising to see such a strong link between the NAW and the PNA sector WRs. Although you mention that previous studies show a similar effect of North Atlantic temperatures on the troposphere in the Pacific sector – could it be that the link partly also acts the other way round, i.e. that the occurrence of certain PNA sector WRs (or a certain ENSO state) affects the SSTs in the North Atlantic and thus the NAW? At least this would be

intuitive from a storm-track dynamical point of view, as the PNA sector WRs strongly influence the entrance of the North Atlantic storm track. Having these strong dynamical links between the four indices and the tropospheric dynamics in mind, I suggest that you either discuss / address this explicitly in your manuscript (by also "weakening" all the causality statements) or, optimally, that you gain some more insight in the causality by doing some kind of linear regression analysis considering time lags (similarly to Section 2.2. in Manzini et al., 2014, https://doi.org/10.1002/2013JD021403), if this a possible approach in your framework. The latter analysis may help to shed some more light on this chicken-and-egg-like problem. Summarizing my comment in other words: I think it is very helpful for further research to include these four indices into your study, but I just think one should treat them more as phenomena that may be strongly and mutually coupled to the WRs themselves.

- Thank you for the comment and the many literature suggestions. We agree on the fact that in Section 4.2 (in its current form) only the link between the "drivers" and the frequency of the regimes is apparent, however the found relationship does not prove any causal link between them. In this respect, we also agree with the reviewer, that the term "driver", without further specification, may be misleading. Therefore, we have decided to add the term "potential" to the title of Section 4.2 ("Potential drivers of future circulation changes"), to highlight this uncertainty, and use more neutral words (link, connection, relationship..) when referring in the text to the correlations found. Also, a comment on the fact that the links may be due, at least partly, to a reversed causal relationship or to an external forcing influencing both processes in a similar way, are included to the discussion in the revised manuscript at lines 392-395: "*This analysis aims at finding significant relationships between the set of potential drivers and the WR frequency trends. Of course, a significant correlation between two quantities does not demonstrate the existence of a causal link - as they might be responding to a common external forcing - nor gives indication on the direction of such link. Nevertheless, this can provide an insight about the interconnection between mid-latitude climate variability and large scale changes in GCMs that deserve further investigation.*"

The definition of the NAW has also been modified, and two different regions in the North Atlantic has been considered for the Atlantic warming in order to better evaluate the relative role of the tropical North-Atlantic (TNAW) and the subpolar gyre/warming hole region (SPGW). We included a new comment in the discussion at lines 430-431, suggesting that it may also be that both the TNAW and the change in the Pacific WRs frequency share a common external driver: "*However, this significant regression may also be produced by a common external driver, influencing both the TNAW and the PNA regimes, like a change in the ENSO forcing (Fredriksen et al., 2020).*"

A thorough analysis is needed to understand whether (or not) the found correlations imply any causal link, however such study goes beyond the scope of the present paper, and it will be carried on in a further study. Thank you for suggesting the method applied in Manzini et al. (2014). Unfortunately, a similar time-lagged correlation analysis is not suited to our case, since most processes involved in the WR transitions have typical lifetimes which are relatively short (usually synoptic). Among all potential drivers considered in Section 4.2, only the stratosphere-troposphere connection has a characteristic time scale of a few months, which is enough for the time-lagged correlations to appear. On the other hand, the dynamical link with the SSTs in the North-Atlantic or the tropical tropospheric temperature is characterized by faster time scales, not suited to that analysis.

Minor comments

• L28-30: I see that several studies find a poleward shift of the upper-level jet caused by the UTW. You additionally mention an intensification of the upper-level jet due to the UTW. However, it is not obvious to me why a stronger meridional temperature gradient in the upper troposphere strengthens the jet on the same level? According to the thermal wind balance, the upper-level jet should primarily be driven by the meridional temperature gradient in the lower troposphere (as, for instance, Hassanzadeh et al., 2014, considers by looking at changes of near-surface meridional temperature gradients on jet intensity). Can you explain this from a dynamical point of view?

We realized that we placed the wrong reference here: as you say, Hassanzadeh et al. (2014) consider the role of the near-surface temperature gradient in affecting the mid-latitude circulation, and purposely exclude the effect of the UTW (mainly driven by increased latent heat release, not present in their dry model). We changed the reference to Barnes and Screen (2013) (https://onlinelibrary.wiley.com/doi/epdf/10.1002/wcc.337), which discusses the influence of the UTW on the jet stream as opposed to the Arctic Amplification (see Section "Tug-of-War: Tropics Versus Poles").
Thermal wind balance relates the vertical wind shear with the meridional temperature gradient. It is true that the wind on a constant pressure surface is not influenced by the meridional temperature gradient at the same height. At the same time this implies that the meridional temperature gradient immediately below the considered isobaric surfaces does contribute to the thermal wind. In our case we define the UTW as the temperature trend in the layer between 400 hPa and 150 hPa, which indeed spans part of the pressure heights located (along the vertical) near the jet stream core, and thus able to have an influence on the jet stream through the thermal wind balance. Moreover, the UTW is a proxy for the warming of the whole tropospheric column in the tropics and is linked with the meridional temperature gradient on a larger vertical interval. This is very clear in Fig. 1 of Shaw (2019) (https://link.springer.com/article/10.1007/s40641-019-00145-8), which shows that the warming in the UTW drives an increase in the meridional temperature gradient in the upper troposphere, that reinforces the jet stream. This is very clear in the Southern Hemisphere, while in the Northern Hemisphere this is moderated by the decrease of the temperature gradient at the surface due to Arctic amplification.

• L32: I would add Pithan and Mauritsen, 2014 (https://doi.org/10.1038/ngeo2071) to the reference of Screen and Simmonds, 2010, who discussed the mentioned "several other positive feedbacks" in more detail.
• L75-76: I would add Michelangeli et al., 1995 (http://shorturl.at/kmKW2) here.

 - Thank you for the suggestions. We added the references in the paper.

• L76: I would reword the sentence "each WR has a different impact on the climate of the downstream region" a bit, because it sounds like WRs are defined in a domain upstream, e.g., over the North Atlantic, to investigate surface weather downstream, e.g., over Europe.

- We rephrased the sentence, which now reads: "each WR has a different impact on the climate of the region considered".

• L76: I suggest to rename the abbreviation for the Pacific-North American sector in the whole manuscript to something like PAC or PNAM, because using PNA becomes confusing later on due to the two equally named regimes PNA+ and PNA- (for instance, at the end of line 214 it is not unambiguous whether you talk about all four PNA sector WRs or only PNA+/-).

- Thank you for the suggestion, it is true that the current notation might be ambiguous. We changed

from PNA to PAC when referring to the Pacific sector.

• L70-85: Could you give a very brief summary (two to three sentences) of previous studies investigating WR changes in GCM simulations / CMIP projections? I think you partly do that later in the results section, but it may be helpful to get an overview of studies here already.

- Most studies of WRs in models focused on the model performance in control/historical simulations (Dawson et al., 2012; Cattiaux et al., 2013; Weisheimer et al., 2013; Dawson and Palmer, 2015; Strommen et al., 2019; Fabiano et al., 2020a).
As far as we know, changes of WRs in CMIP5 projections were only analyzed by Cattiaux et al. (2013, doi: 10.1007/s00382-013-1731-y) and by Ullmann et al. (2014, doi: 10.1002/joc.3864) for the EAT sector, which however used a substantially different approach and found different results for the regime frequency change (a stronger NAO- increase and no significant change, respectively).
We added references to these works in the introduction at lines 79-82: *"Many works in literature studied how WRs are reproduced by GCMs, but mostly focused on the model performance in control or historical simulations (Dawson et al., 2012; Cattiaux et al., 2013b; Dawson and Palmer, 2015; Weisheimer et al., 2014; Strommen et al., 2019; Fabiano et al., 2020). Changes of WRs in CMIP5 projections were analyzed by Cattiaux et al. (2013a) and by Ullmann et al. (2014) for the EAT sector."*
We also added a comparison of our CMIP5 results with the Cattiaux ones in the discussion section at lines 326-328: *"The change of WR frequency observed for RCP8.5 is consistent with the result by Cattiaux et al. (2013a), although they observe a larger increase of NAO-, which may be due to a different treatment of the climatological mean state."*

• L116-118: Can you elaborate a bit more on why it is necessary to detrend the historical data with the described approach before identifying the WRs? More specifically: How robust / meaningful is the described linear trend in the Northern Hemisphere area-averaged geopotential height, considering for instance the substantial multi-decadal variability in the large-scale circulation over these 50 years? Could it be that the trend (and thus the WR identification) becomes significantly different when considering, for instance, only the last 40 instead of 50 years (which is often done when investigating the ERA-Interim period only)? How reasonable is it to detrend, for instance, the North Pacific with a linear trend that is obtained from an area average over the whole Northern Hemisphere (including the North Atlantic)? Does your WR pattern identification change if you do not detrend the historical data (which is often done in other studies to my knowledge)?

- A new figure (S11) was added to the supplementary material and is reported below. The figure (left panel) shows the average geopotential height at 500 hPa (in units of meters) for all models in the historical period and ssp585 future scenario. The need for the polynomial detrending is clear from the scenario averages, which show a non-linear behaviour. When performing the scenario detrending, we judged more correct to also detrend the historical data with a linear term, since the increase in the geopotential height is already clear for the 60s to the end of the historical period. The historical increase reaches up to 20-30 m for individual models, which can certainly change some daily assignments, possibly slightly changing the frequencies and creating spurious frequency trends in the historical period.
The choice to calculate the trends on the whole Northern Hemisphere was initially motivated by avoiding influences on the trend due to decadal basin-wide fluctuations, such as the AMV, which would affect less the hemispheric trend. Nevertheless, we observed that the difference in the historical and future trends when considering the whole hemispheric or the sectorial averages is very small (see right panel).

[Figure]

[Figure]

Figure 1. Average geopotential field in the Northern hemisphere (30-90N) for all model simulations in the historical+ssp585 scenario (scatter) and the linear/polynomial fit for the historical/scenario respectively (lines). Right panel: Detrending for the ssp585 scenario simulation with EC-Earth3, considering different areas for the average: NH (0-90N), Arctic (70-90N), NH (30-90N), EAT and PNA as in the paper, rus (30-90N, 40-140 E).

• L122-123: If I understand correctly, you ultimately apply the EOF analysis to unfiltered daily Z500 anomalies, right (apart from the running mean climatology you subtract)? What is your idea behind using the daily anomalies like this, without applying any low-pass filter beforehand? Would the latter change your result – did you test this?

- Yes, we apply the EOF analysis to the unfiltered daily anomalies, as it has been done in many other works in literature (e.g. Cassou, 2008; Dawson, 2012; Cattiaux, 2013; Strommen, 2019). Other works did apply a 10-day Lanczos filter (Straus, 2007; Dawson, 2015; Madonna, 2017), but the regime patterns obtained there are not significantly affected by such operation. We also checked this on ERA data and found no significant shift in the regime centroids when applying the filter. However, applying a low-pass may have an impact on the daily assignments and we judge the no-filtering approach to be more conservative in this sense.

• L123-127: Is it correct that you calculate the future Z500 anomalies (used for detecting the WRs) by subtracting the Z500 climatology (or the seasonal cycle, as you call it) based on the historical period and not based on the future period? Is this what you mean with the last sentence in this paragraph? I think this is crucial but may not be fully clear from the text.

- Yes, this is correct. Thank you for the comment, we rephrased the paragraph to make this clearer (now at lines 130-134): "*The seasonal cycle is computed averaging the data day-by-day at each grid point and applying a 20-day running mean to remove higher frequency fluctuations. However, the seasonal cycle computed in the historical simulations generally differs from the seasonal cycle found in scenarios. Since these differences are part of the change in the midlatitude circulation, it is important to take them into account. Therefore, for each model and scenario, the mean seasonal cycle is computed in the reference period of the corresponding historical simulation (1964-2014 for CMIP6, 1964-2005 for CMIP5).*"

• L129-131: Here it would be worthwhile citing some other studies using similar regional domains

for the EOF analysis.

- We added Dawson et al. (2012) and Weisheimer et al. (2014) as references for the EAT and PAC cases.

• L131-136: I am not sure if I fully understand the procedure: Do you apply an EOF analysis both on the observed anomalies (to get the 4 observed PCs), and also separately on the modeled anomalies (to get the 4 pseudo-PCs) to calculate the "computed regimes"? In the current form, it sounds like you apply the EOF analysis only on the observed anomalies, and you then project the modeled anomalies into this (observed) phase space, as a basis for both the "computed" and "projected" regimes.

- The EOF analysis is in fact applied only to the observed anomalies, and the space spanned by the 4 reference EOFs is then used as a reference phase space for all model simulations. So, for the model simulations, the daily anomaly field is directly projected on the reference EOFs, to obtain the series of pseudo-PCs. This procedure was first adopted by Fabiano et al. (2020a) and additional comments and sensitivity tests can be found there. The advantage of considering a single reference space is to allow a direct comparison of the cluster centroids from different simulations in a consistent way. We rephrased the paragraph to make this clearer (now at lines 142-144): "*The phase space spanned by these EOFs (hereafter "reference phase space") is then used for both the reanalysis and all GCMs simulations: all anomalies are projected onto this reference phase space, obtaining the 4 leading Principal Components (PCs) for the reanalysis dataset and 4 pseudo-PCs for each model simulation.*"

• L146-148: Out of curiosity, did you calculate the "computed regimes" also for the future simulations? If yes, are they different, and do they tell us something about changes in modes of variability in the large-scale circulation?

- Unfortunately, we have not yet calculated the computed regimes for the future simulations. We agree that this would be a really interesting point to consider, but we expect the effect to be of second-order with respect to the change in frequency, and probably more difficult to assess. For this reason, we decided to focus only on the regime frequencies in this work, and leave the study of potential changes of the modes of variability to a future work.

• L160-162: Could you briefly mention in the manuscript whether a higher or lower variance ratio is generally desirable for a WR definition (independent of the comparison between observations and model)? I guess a WR definition is "better" (i.e., the WRs are more distinct) the larger the variance ratio is, because a high variance ratio implies a relatively large distance between the cluster centroids compared to the distances within a centroid, right?

- Yes, a higher variance ratio is desirable in cluster analysis. However, the comparison with the observation is key here, since models tend to have too low variance ratio on the EAT sector and too high in the PAC one. The Pacific regimes should not be so clearly defined in models, and this reflects some misrepresentation in the midlatitude circulation that needs further study. We added a clarification at lines 170-172: "*In cluster analysis, a larger value of this ratio is generally desirable, indicating that the clusters are well separated from each other. For WRs, the distance from the observed variance ratio is an indicator of the overall model performance in simulating the regime dynamics (Fabiano et al., 2020).*"

• L195-196: Out of curiosity, do you know whether there are preferred circuits / transitions between the PNA sector WRs, considering the fact that they resemble (different states of) Rossby wave trains originating from the tropics?

- Thank you for the interest. Unfortunately we have not yet carefully studied the transition probabilities between the different PAC states. However, we computed the transition probability for the reanalysis data. At first sight, the highest transition probability from PT is towards the PNA-regime, consistent with the Rossby wave train view. For all other regimes, the transition to PT seems favoured.

• Figure 2: I really like the way you compare the WR representation / biases in Figure 2! I think it could be of interest for the CMIP community to additionally see in the Taylor diagram how individual model centers improved (or worsened) their WR representation between phase 5 and 6. Could you use a specific symbol (instead of a dot for every simulation) for the same model (or for related model simulations / centers)? Or does this make the figure too overloaded?

- Given the large number of models considered, we thought that the figure would have been simpler with just the model ensemble information. However, we produced an analogous figure with specific symbols for each model, and added it to the Supplementary material (Figure S1). Nevertheless, the model centers represented in the two CMIP phases differ, so the check for the improvement of a particular one will not be possible for all of them.

• L212-214: I kind of see your argument of an overall improvement between CMIP5 and CMIP6 visually, but can you "proof" this with a certain measure of significance (considering the relatively small number of model simulations)? Is this the degree of overlap of the shaded blue and red ellipses (if yes, please specify in the manuscript)? I would also be careful with concluding that the two NAO WRs improve more than the others – this is not that convincing considering the large inter-model spread for instance in the NAO+. Also, in principle it can be that the same model center does not improve but rather worsen from phase 5 to phase 6, which would become visible if the symbols were changed as suggested in the previous comment (for instance, the very top-left red dot in the AR diagram probably indicates such a case). Can you elaborate a bit more on this in the manuscript and, in case you do not indicate the individual models with a symbol, say whether all (or most) models improved from phase 5 to phase 6?

- Yes, the degree of overlap gives a measure of the significance of the improvement from CMIP5 to CMIP6 ensembles, we added a comment on this in the text and removed the claim on the two NAO regimes improving more than the other regimes. Lines 225-227 are changed accordingly: "*For the EAT sector (Figure 2, first row), the CMIP6 ensemble shows an improvement with respect to the CMIP5 counterpart for all regimes, although the intermodel spread is quite large and the ellipses significantly overlap (apart for NAO-).*"
Your suggestion of looking at the change in the metrics for individual model centers, and see how many of them improve between the two phases, would certainly be a good way to assess this more quantitatively. We provided an indication of how many of the "matching" models improve from CMIP5 to CMIP6 in the pattern correlation, variance ratio and frequency bias in Table 2. The results confirm those of the main analysis, with most "matching" models improving from CMIP5 to CMIP6 for all metrics. Weaker improvements are found for the EAT regime frequency and for the PAC pattern correlations, as was also seen in the main analysis. Lines 244-245 now read: "*The results shown in Figures 2 and 3 are confirmed when looking at the performance of models developed by the same institution in the two phases, reported in Table 2: most models improve from*

*CMIP5 to CMIP6 for all three metrics."*

• L212-214: The clearly smallest inter-model spread and generally smallest bias in the NAO- WR in the Taylor diagram may indicate a higher intrinsic predictability of the NAO- WR compared to the others (also compared to NAO+). Does this make sense, and did you think about this? And, if yes, do you have an explanation for this, or do you know whether this has been shown before?

- The NAO- mean regime pattern is known to be the best reproduced in historical model simulations (see for example Strommen et al. 2019, Fabiano et al. 2020a). However, we never considered the possible connection with the regime predictability. On one side, the fact that a simulation correctly represents the observed pattern has no clear implication for its predictability. However, it seems plausible that a correct representation of the regime pattern might be due to a better representation of the processes behind the onset and persistence of the regime, which in turn could also give a better skill in predicting the regime itself. We are not aware of studies showing a better predictability for the NAO- though, but we will keep this suggestion in mind when analyzing seasonal simulations.

On the other hand, the PNA sector WRs generally seem to be harder to capture properly, considering the large inter-model spread. Do you have an explanation for this? Could it be related to the strong dependence on the tropics, implying that models with a bad representation of the tropical-extratropical interaction perform substantially worse in terms of PNA sector WRs? I know this is beyond the main focus of this study, but I think it would be helpful to briefly discuss these aspects and speculate about possible reasons in a few sentences.

- We agree that the PAC regimes are generally more difficult to capture than for the EAT sector. This may reflect a larger natural variability in the regime structure, as suggested by the smaller values of the variance ratio of the reanalysis on the PAC sector. The link with the tropics might be key to that, since these regimes are strongly influenced by the ENSO forcing. A different representation in terms of amplitude and frequency of ENSO events in the different model simulations could perturb the regime patterns in different ways, leading to a larger spread. In this regard, there is some indication that the model performance for the EAT sector might be linked to the SST representation (see Fabiano et al, 2020a). It would be interesting in this sense to analyze the observed natural variability in these regimes, and compare it to that observed in the EAT sector. We added a brief comment on this in the paper at lines 231-233: *"PAC regimes appear to be more difficult to capture than EAT ones. This may reflect a larger natural variability in the observed regime structure, as suggested by the smaller variance ratio of the reanalysis for PAC with respect to EAT. Also, the PAC regime patterns might be influenced by the specific history of each model simulation in terms of amplitude and frequency of ENSO events."*

• Figure 3: I understand the idea of Figure 3 from a perspective of condensing information, but I do not see the scientific reason for plotting the frequency bias against the variance ratio because there is no direct link between the two. Hence, it could confuse the reader because a linear relationship may be expected by this way of plotting. If there is a link, please clarify in the text. Otherwise, I suggest showing two vertical box-whisker plots, one for the frequency bias and one for the variance ratio (with a horizontal black line for the corresponding ERA-Interim value). Furthermore, please indicate the units for the frequency bias.

- The idea of the figure was just to synthetically show two important metrics "at once", but that might not be the best way to do this. Indeed, there is no direct link (that we know of) between the two quantities and we accept your suggestion to change this figure into a box-whisker plot in the

revised manuscript.

• Figures 2 and 3: How would you relate the biases shown in your figures to the well-known blocking biases in GCMs (e.g., Davini and D'Andrea, 2020, https://doi.org/10.1175/JCLI-D-19-0862.1)? Do we also see this somehow in your figures?

- In the EAT sector, there is a strong link between regimes and blocking events (see Madonna et al. 2017 and Figure 10 in Fabiano et al. 2020a). In particular, most of the blocking events in the EAT sector coincide with NAO- and SBL states, with a lesser contribution of AR. Models generally are able to reproduce this correlation, but generally struggle to reach the observed intensity of the signal. Apart from the model biases in catching the regime-blocking connection, it seems plausible that a smaller frequency bias might indicate a smaller bias in the blocking representation. Also, a link between blocking bias and variance ratio has been observed in a multi-model ensemble in Fabiano et al. (2020a). As for the PAC sector, although we are not aware of a similar work linking regimes and blocking events, it is very likely that the Bering Ridge is associated with the North Pacific blocking.
We added a brief comment on this at line 240: "*Given the strong link between WRs and blocking events (Madonna et al., 2017; Fabiano et al., 2020), the reduction of the WR frequency bias is in line with the smaller biases in the blocking frequency observed for CMIP6 models (Davini and D'Andrea, 2020).*"

• L222-225: The overall different variance ratios between the PNA and EAT WRs compared to observations are indeed interesting. In the case of the PNA sector WRs, however, a higher than observed variance ratio does not necessarily mean that the models perform better than for the EAT WRs, right? It just means that the models distinguish more (too?) strongly between the different WRs. So I do not really understand this statement here, because, regarding my previous comment (about the larger spread in the Taylor diagram for the PNA sector WRs), I would rather think the models overall have more problems for the PNA sector WRs. . .

- Yes, the representation of the PAC regimes in models overestimates the variance ratio. We agree that the sentence has not been formulated well. Actually, our hypothesis here is that the tropically-induced modulation of the North Pacific regimes might be too strong in models. Therefore the regime structure turns out to be too "deep" and there is less room for larger deviations from the attractors. Molteni et al. (2020) showed that the response of the NAO index to the tropical Pacific forcing is well represented in models, while the teleconnection of the Atlantic sector with the Indian Ocean is not well caught. However, this does not really help our argument here. We changed the sentence in the revised manuscript (line 237-239) as follows: "*It is worth noting that – opposite to the EAT sector – models tend to produce larger variance ratios for the PAC regimes than it is observed, which might be due to an excess in the tropically-induced modulation of the PAC regimes in models.*"

• L230-233: I find it somehow surprising that the model biases in reproducing the observed regimes are smaller for projected regimes than for computed regimes. Can you explain? Is it simply because applying the k-means clustering to each model simulation (i.e., the computed regimes) yields slightly different and thus "new" regimes, hence, you kind of compare apples with oranges in the Taylor diagrams in Figure 2?

- The computed regimes of each simulation are reordered as to best match the observed series. Generally there is a quite good one-to-one match, so we are still comparing apples with apples in most cases. However, the natural variability of the system is large and the k-means on a relatively

short timeseries (50 years) can produce, in some cases, centroids that are shifted in phase space. Large differences were also observed between different ensemble members of the same model (Fabiano et al., 2020a), with occasionally some "rotated" regimes being produced and correspondingly very bad performance. This might be the case for the few outliers that can be seen, for example, for the PNA- pattern, with spatial correlations close to 0. This does not happen for the projected regimes since we force the cluster centroids to the observed ones and the regimes, along with their dynamical implications, are always well defined. Indeed, the pattern biases result smaller for the projected regimes (Figure S1). However, it is less obvious that the frequency and variance ratio biases should be smaller as well. Nevertheless, this is what we observe in Figure S3, hinting that the K-means might enlarge some biases by misplacing the cluster centroids, while the "real" attractors might be closer to the observed ones.

• Figures 4, 6, 7: Considering the relatively small number of models, I wonder how robust the distributions in the box-whisker plots are. Did you check whether certain distributions are skewed due to, for instance, a clustering of several model simulations from the same model center? I guess the Welch's t-test does consider that particular problem. Nevertheless, it may be helpful for the reader to replace the box-whisker plots with violin plots additionally indicating the density within the distribution.

- The main result we want to highlight with Figures 4, 6 and 7 is the shift in the WR frequencies under increasing greenhouse forcing. The box-whisker plots are indeed enough and clear to show the multi-model ensemble mean. Besides, some information on the spread and the skewness of the distribution can be gained. We prefer not to change the plots to violin plots to avoid overloading the figure and possibly make the main result more difficult to read. Nevertheless, the equivalent Figures with violin plots are available in the Supplementary Materials as Figure S5 and S8.

• Figures 4, 6, 7: I would color the historic box in black (and all the future scenarios in color, as it is), just as a suggestion.

- Thanks for the suggestion. We changed the color of the historical box to black to help distinguish it from the scenarios.

• L257-263: The apparently non-linear response of the NAO- to the CO2 forcing is very interesting! Furthermore, the temporal development in the different simulations in Figure 5 shows an interesting multi-decadal variability. You mention that this will be analyzed in further studies. Can you nevertheless speculate about some potential reasons? Could it be related to some kind of tipping points in external forcings such as the Greenland ice sheet (which could affect the Greenland high) or sea surface temperature?

- The most promising hypothesis is related to the aerosol forcing, that could have a role in driving in-phase AMV oscillations in the model simulations. This has been hypothesized for the observed AMV (see e.g. Zhang et al. Have aerosols caused the observed Atlantic Multidecadal Variability? J. Atmos. Sci. 70, 1135–1144 (2013); Qin et al. 2020, DOI: 10.1126/sciadv.abb0425). In turn, the AMV perturbs the observed frequency of the NAO+/- regimes (a positive AMV increases the NAO- frequency). It is not clear whether a similar process might be at work for the future scenario period, but the way seems promising. We added a brief comment on this at line 283: "*These might be related to the aerosol forcing, which has been hypothesized to have driven the observed AMV*

*oscillations (Zhang et al., 2013; Qin et al., 2020). However, it is not clear whether a similar process might be at work for the future scenario period and further analysis on this topic will be carried out in a different study.*"

• L264: How does the temporal evolution for the PNA sector WRs in the CMIP simulations look like (analog to Figure 5)? Does it also exhibit any multi-decadal variability in specific WRs? I would suggest adding this figure to the supplement.

- The figure has been added to the supplementary material (Figure S10). The temporal evolution for the PNA sector shows some multi-decadal variability, in particular for the PT regime (for which a minimum is found at the end of the historical period) and for the PNA+/PNA- scenarios. Interestingly, the RCP8.5 scenario of CMIP5 deviates from the CMIP6 projections for almost all regimes. However this is in part due to the plot construction and to differences in the historical mean frequency, since the differences with respect to the CMIP6 historical ensemble are shown here.

• L277-289: Thinking in terms of WR life cycles, the strong correlation between changes in WR frequency (Figures 4, 6) and WR persistence (Figure 7) implies that there does not seem to be changes in numbers of life cycles but rather changes in the duration of individual life cycles (which ultimately make the changes in WR frequency). Is that correct? If yes, can you discuss this with a few sentences in the manuscript? It could also be interesting / helpful to plot changes in WR frequency against changes in WR persistence. Depending on the robustness, this finding may to some degree also have implications for the (operational) predictability of WRs for instance on subseasonal-to-seasonal time scales.

- Thank you for the suggestion. We added Figure 10 to the manuscript, which shows the change in the average number of regime events per 100 days. Also, we added lines 311-315: "*Figure 8 shows the number of regime events per 100 days. The changes in the regime frequencies might be seen as the combined effect of the changes in the regime persistence and the changes in the number of regime events. For the EAT sector, both have a comparable role in the frequency change of AR and SBL, while the increased persistence seems the main factor in the NAO+ change. For the PAC sector, the increase in PT frequency is driven by longer persistence, despite no significant change in the number of events, while the opposite is true for the change in the BR regime frequency.*"

• L310-311: What are the projections for future ENSO occurrence? Do we expect (significant) changes? Can you cite some of these studies here?

- A recent study of ENSO occurrence CMIP6 projections (Fredriksen et al., 2020, 10.1029/2020GL090640) shows a tendency for an increase of ENSO variability under global warming, and interestingly this change appears to be related mostly to an increase in positive El Niño events. This is in line with our hypothesis that the PT regime frequency increase might be related to a stronger tropical forcing, we added a comment on this in the discussion at line 347: "*A recent study of ENSO occurrence in CMIP6 projections (Fredriksen et al., 2020) shows a tendency for an increase of ENSO variability under global warming, and interestingly this change appears to be mostly related to an increase in positive El Niño events.*"

• Figure 8: Does NML stand for the hemispheric zonal mean?

- NML stands for Northern Mid-Latitudes, which indicates the region from 30N to 90N, used to

calculate the trends. However, the term might not be really adequate, since the region includes the higher latitudes as well. We changed this to NH (30-90N) in the revised manuscript.

• Figure 9: Just for clarification, does the shading in this figure show the mean Z500 (grid-point level) in the future simulation minus the zonal mean (at every latitude) shown in Figure 8? How does Figure 9 compare to a map that simply shows the future mean Z500 minus the historic mean Z500 (both on a grid-point level)?

- The figure shows the residual trend at each grid point in the Z500, after removing the zonal trend. Analogously, Figure 8 shows the residual zonal trend, after removing the global NH trend. The difference of future and historical Z500 is dominated by the global NH trend, which is largely positive, so it would be difficult to discern changes in the stationary eddies from that figure. We clarified this in the captions.

• L344-350: How sensitive is your analysis to the latitude / pressure boundaries used to define the four metrics?

- Apart from the North Atlantic warming (NAW), the other metrics had already been used in other works (e.g Oudar et al., 2020; Peings et al., 2018; Zappa and Shepherd, 2017). We used here the same pressure/latitude boundaries as defined in Oudar et al. (2020), very similar to those in Peings et al. (2018), but a different choice (as in Zappa and Shepherd, 2017, that consider individual pressure levels) is not expected to change the results dramatically. Unfortunately, we do not have a quantitative estimate on this. As for the NAW, we split it in two better defined subregions (tropical Atlantic and subpolar gyre), in order to evaluate their relative contributions.

• L362: Please add Figure S7 to the supplement, because it's missing.

- Thank you for pointing this out. The new figure has been added to the supplementary as Figure S12.

• L366-376: I would add a reference to, e.g., Ambaum and Hoskins, 2002 (http://shorturl.at/ipDO9), who proposed a mechanism for the strong NAO-polar vortex coupling (see previous comment).

- Thank you, we added the citation to the text.

• L388-401: The changes in WRs in a future climate must be strongly linked to changes in extratropical cyclone activity and thus the storm track. Can you briefly discuss or at least speculate here whether and how some of your results (e.g., the increase in NAO+ frequency) might relate to the expected changes in extratropical cyclone frequency, location, and intensity? I assume this question is more complex than we think, but it would be nice to at least mention these questions in the conclusions and thus make a bridge toward the cyclone research community. Because in the end, changes in WRs are also a result of changes in cyclone activity. . .

- We added a couple of comments on this in the discussion at lines 329 and 342:
*"The strong increase in the NAO+ regime frequency is in line with the change of storm track activity in CMIP6 projections analyzed by Harvey et al. (2020), which shows an intensification of*

*the activity over the North-Atlantic and central/northern Europe, with a center on the British Isles and an increased penetration of perturbations into the continent. A corresponding decrease of perturbations at very high latitudes is also in line with a decrease in the AR regime, which tends to push the jet poleward."*
*"Also, a strong decrease of the storm track activity under SSP585 has been observed over the whole Northern American continent, and an increase in the central North Pacific (Harvey et al., 2020). This agrees well with the prediction of an increased PT regime, that blocks the entrance of perturbations in the continent."*

• Throughout the manuscript, you often write certain phenomena with capital letters, which I would not do. For instance, change "Weather Regimes" to "weather regimes", "Polar Stratospheric Temperature" to "polar stratospheric temperature", "Polar Vortex Strength" to "polar vortex strength" etc.
• You misspell "Pacific Through" (instead of "Pacific Trough") several times in the manuscript (including the Abstract)
• L11: Change to "A major challenge for the climate community is to understand how a warmer climate . . ."
• L13: Change to ". . . inextricably related to regional impacts . . ."
• There are a few further grammatical inconsistencies throughout the manuscript, which should be detected when carefully revising the manuscript.

- Thank you for the language corrections, we implemented them in the revised version.

We thank Reviewer 2 for the constructive comments to the manuscript, which stimulated a deeper understanding.

This paper deals with the changes in weather regimes in the Euro-Atlantic and Pacific-North American sectors, in both CMIP5 and CMIP6 models. A comparison of simulated and observed weather regimes is done and then the changes in frequency and persistence of the weather regimes are characterized. Potential drivers of those changes are finally discussed. This is a nice paper, well written, with interesting results. The paper is concise, despite the large amount of work necessary to do these analyses. I recommend publication of the paper after some moderate revisions.

Main comments

The weakest part of the paper (but also, potentially, the most interesting one) is probably section 4.2. The analyses themselves are interesting but I think the interpretation of the results should be more cautious. It is not because a significant (?) correlation between changes in weather regimes frequency and what the authors call the "drivers" is found that a causal relationship exists. The weather regimes could themselves impact these drivers, and other factors, not studied by the authors may physically explain both the changes in weather regimes and in the drivers. More drivers could also have been considered: e.g. many studies have shown the potential impact of SST anomalies (especially in the Tropics, but not only) on large scale circulation, including on weather regime occurrence. I think it is difficult to discuss the potential drivers of weather regime changes without investigating the role of regional SST changes.

- Thank you for the comment. We agree with the reviewer that the term "driver" might be ambiguous because it suggests a causal relationship, which cannot be directly implied by our analysis. However, our choice has been somehow inspired by recent literature (e.g. Zappa & Shepherd 2017, Oudar et al. 2020), where processes that can potentially affect the mid-latitude circulation under increased forcing are referred as "drivers". In response to this comment and to a similar argument by Reviewer 1, we added the term "potential" to the title of Section 4.2 and referred to the correlations found as "link/relationship/connection".
Also, a comment on the fact that the links may be due, at least partly, to a reversed causal relationship or to an external forcing influencing both processes in a similar way, are included to the discussion in the revised manuscript at lines 392-395: "*This analysis aims at finding significant relationships between the set of potential drivers and the WR frequency trends. Of course, a significant correlation between the two quantities does not demonstrate the existence of a causal link - as they might be responding to a common external forcing - nor gives indication on the direction of such link. Nevertheless, this can provide an insight about the interconnection between mid-latitude climate variability and large scale changes in GCMs that deserve further investigation.*"
Also, we included a new comment in the discussion at lines 430-431, suggesting that it may also be that both the TNAW (tropical North-Atlantic warming) and the change in the Pacific WRs frequency share a common external driver: "*However, this significant regression may also be*

*produced by a common external driver, influencing both the TNAW and the PNA regimes, like a change in the ENSO forcing (Fredriksen et al., 2020).*"

To assess the role of regional SSTs changes, we investigated the correlation patterns between WR frequency trends and changes in the surface temperature (tas) across the multi-model ensemble, for the ssp585 scenario. This has been done calculating the correlation between the tas trends of all models at each point of a common grid and the regime frequency trends of all models. Both the tas and frequency trends were divided by the global temperature trend before computing the correlations. For the EAT regimes no significant pattern appears, except for the NAO+, which shows significant correlations in some areas of the central and western tropical Pacific. The situation is more interesting for the PAC regimes, which show significant correlations in the tropical North Atlantic and in the North Atlantic subpolar gyre. Indeed, the North Atlantic warming (NAW) was considered as a potential driver in Section 4.2. However, we decided to split the NAW in two parts, in order to assess the relative importance of the tropical North Atlantic (TNAW) and the North Atlantic suppolar gyre (SPGW). In addition, significant correlations for the PNA+/-regimes are also found in Northern Africa, the Indian ocean, the Northern Pacific and some smaller regions in the Southern ocean. However, the understanding of these correlations requires further analysis that goes beyond the scopes of the present paper.

I also think that a discussion on the potential causes of the differences seen between CMIP5 and CMIP6, which are sometimes quite large, should be added to the paper. An interesting analysis would be to evaluate whether differences between the changes in weather regimes are linked to differences in the changes of the drivers between CMIP6 and CMIP5 (if the authors are right about the drivers, it should be the case).

- The differences between the changes observed in RCP85 and ssp585 might be mostly related to differences in the forcing. In fact, in terms of $CO_2$ concentration, the CMIP5 RCP85 scenario represents an intermediate narrative between CMIP6 ssp370 and ssp585 (Meinshausen et al., 2019; Tebaldi et al. 2020, https://doi.org/10.5194/esd-2020-68). Moreover, a recent study with the EC-Earth model finds that about half of the difference in warming by the end of the century when comparing CMIP5 RCPs and their updated CMIP6 counterparts is due to difference in effective radiative forcings at 2100 of up to 1 Wm-2 (Wyser et al., 2020; doi:10.1088/1748-9326/ab81c2). Also, the models considered for the CMIP5 scenario differ from the CMIP6 ones.

Many methodological choices are necessary in a weather regime analysis: e.g. preprocessing through EOF analysis or not, if yes number of principal components retained, precise algorithm (many variants of k-means exist), choice of the number of clusters, whether all the days are classified or not, if not what is the criteria for not classifying some days, which variable is used (slp, zg), how to best take into account the mean increase in zg due to global warming etc. I somewhat understand why the authors don't want to discuss all these aspects, show sensitivity tests etc. The paper would be too long and less interesting. But nevertheless some aspects deserve to be better justified, at least with a sentence. Sensitivity tests could also be added in SI (as the authors decided to add SI to the paper). For example, the authors have chosen to first apply an EOF decomposition to zg and then to retain the 4 leading EOFs. They give no justification for this pre-processing step. The use of principal components for cluster analysis has been common in the past but I suspect that in many old studies, it was more a question of dimension reduction in order to run classification algorithms on computers with limited resources. The use of principal components is also sometimes intended as a kind of filter. If it the case here, the authors should explain their objective(s). It is also unusual to retain such a small number of principal components for clustering, and therefore such a low level of explained variance. I would like to see a discussion on this point and potentially the

result of a sensitivity test: e.g. changes in weather regimes without EOF analysis, or with enough principal components to keep at least 90% of the variance etc.

- We totally agree with the reviewer that many methodological choices are necessary for the Weather Regimes analysis. Excluding the detrending and the idea of *projected* regimes, which here were necessary to take into account the huge (transient) changes in mean climate in the different scenarios, most of the choices adopted in this paper build on Fabiano et al. (2020a). In that paper, a more detailed methodological analysis and discussion are presented, regarding: a) the impact of different number of EOFs; b) changes in the selected region; c) changes in the construction of the climatology; d) the impact of the projection on the reference phase space to obtain pseudo-PCs. In particular, the changes in the Weather Regimes when considering for example 10 EOFs instead of 4 turn out to be negligible for the EAT sector (Section 3.1 in Fabiano et al, 2020a). The same was found for the PAC regimes by Straus et al. (2007), for the case with no filtering applied. This is due to the fact that the regimes are large-scale patterns, well explained by the first 4 EOFs.

Finally, it would also be interesting to characterize the impact of internal variability on the changes in weather regime occurrence, as signal-to-noise may be small regarding atmospheric circulation changes. It would help to better evaluate the importance of future changes, of the differences between CMIP6 and CMIP5, and between scenarios etc. There are a few CMIP6 models with at least 10 members for which Zg is provided (for future scenarios). It would be very interesting to use the members of these models to characterize the impact of internal variability (for a single scenario, ssp585 for example, this is sufficient).

- Thank you for the suggestion, we agree that this would be a very interesting addition. However, the computational and additional storage resources necessary to collect and analyze the daily geopotential height field from all available single-model ensembles is non negligible, considering also that overall we have already analyzed more than 150 daily datasets.

To take into account this issue, we estimated the impact of the internal variability on our results by looking at two model ensembles for the historical and ssp585 simulations: MPI-ESM1-2-LR (10 members) and UKESM1-0-LL (4 members). The results for the EAT sector are shown in Figure 1 below: the running mean of the ensemble mean frequency, with ensemble spread (left panel), and the future change in WR frequency, with ensemble spread (right panel). The variability inside each model ensemble is smaller than that observed for the multi-model ensemble (Figure 4 of the manuscript), or at most of the same order, thus giving confidence in the results obtained using a single member for each model.

[Figure]

Figure 1. Estimation of the impact of the internal variability on the results, based on two multi-member ensembles. Left panel: running mean of the ensemble mean WR frequency for the two models, with indication of the ensemble spread (fill, standard deviation). Right panel: analog of Figure 4 of the manuscript showing the variability of the change in the WR frequency inside the ensemble.

Specific comments

L15. Please define precisely what is meant by "low-frequency" in the paper.

- We intend low-frequency as the variability on scales longer than about 5 days, which in the EAT sector is mainly related to latitudinal shifts of the jet stream. This was clarified in the revised manuscript at line 15: "*The wintertime mid-latitude climate in the Northern Hemisphere is primarily influenced by the low-frequency variability (at timescales longer than 5 days) related to the strength and position of the eddy-driven jet stream.*"

L103. I think that Table S1 with the models and members used should be in the main document.

- Thank you for the suggestion, we moved the table to the main text (Table 1).

L105-108. I'm not sure it is good idea to combine the results of two reanalyses (ERA40 and ERA Interim). Moreover, both of these reanalyses have now been superseded by ERA5, with major improvements in ERA5. I understand that ERA5 is not (yet) available before 1979, but I'm not sure why the authors absolutely need to start in 1964. They could use the period 1979-2015. It would lead to a smaller sample, it is true, but I also think that the hypothesis of a linear trend (section 2.2.1) may be more reasonable on a shorter than on a longer period.

- The use of ERA5 would have been preferred if the whole period had been available in time for this analysis. The need for covering a longer observed period is dictated by the fact that the internal decadal variability in quantities like the WR frequency is quite large, and we evaluated that a period of 50 years would be necessary to assess significant changes. As it is reported at line 186 of the revised manuscript, the variability of the frequencies on this period is estimated to be around 1.6%, which is satisfactory for our scopes. We made an exception to this choice only for the historical simulations from CMIP5, which stops at 2005 by construction and therefore spans only 40 years instead of 50.

In this regard, the combination of the ERA40 and ERAInterim datasets might be seen as a standard workaround, already adopted in several works analysing mid-latitude variability (e.g. Schiemann et al, 2017; Davini and D'Andrea 2020). In particular Dawson et al. 2012 used a combination of these reanalyses to compute Euro-Atlantic clusters, using a methodology very similar to that presented here and showed that the cluster patterns computed using the NCEP reanalysis are almost identical (see Table 1 in Dawson et al. 2012). We have also checked the ERA40-ERAInterim combined dataset vs NCEP on the period considered and found almost no-difference in cluster centroids, with a pattern correlation of about 1.

L118. Please explain (in the paper) why you need to "detrend" the data (by the way, I'm not sure that detrend is the good word considering what is done). And please justify why you use the average on the Northern Hemisphere (30-90N), rather than the averages on the domains of classification, as it is usually done. I think it may have some non-trivial implications, for example if the mean increase in geopotential height, largely controlled by warming, is different between the Euro-Atlantic and Pacific-North American sectors, as it may favor artificially some weather regimes in both domains.

- Thank you for the comment. The figure below (left panel) shows the average geopotential height at 500 hPa (in units of meters) in the Northern Hemisphere (30-90N) for all models, merging the historical and scenario simulations. The need for the detrending is due to the fact that anomalies associated with the WRs are of the order of 100 meters, which are comparable to the average increase in the mean geopotential height field seen in the scenarios. Even if to a lesser extent, also the historical simulations show such an increasing trend. A method to remove the trend - in order to not influence the regime detection has been therefore developed: a linear detrending has been applied to the historical trend and a polynomial detrending has been applied to the scenarios - in order to take into account the fact that scenarios exhibit a non-linear behaviour. In order to retain decadal basin-wide fluctuations, such as the AMV (which we do not want to remove), we decided to calculate the trends on the whole Northern Hemisphere, and not for the separate domains. The difference in the future trends when considering the whole hemispheric or the sectorial averages is very small (see right panel in Figure 3). On the other hand, this choice may have a larger effect on the historical trends, which are calculated on 50 years only (compared to 85 for the scenarios). However, following the above-mentioned argument on the decadal basin-wide fluctuations, the evaluation of the hemispheric trends is more reliable than the equivalent one in sectorial regions.

The two figures have been added to the supplementary material (Figure S11) and this point has been now clarified in the text at lines 123-126: "*We chose to calculate the trend on the Northern Hemisphere (30N-90N) - and not on the separate EAT and PAC domains - in order to retain possible decadal basin-wide fluctuations. Anyway, the difference in the future trends when considering the whole hemispheric or the sectorial averages is very small (see right panel in Figure S11).*"

[Figure]

Figure 2. Left panel: average geopotential field in the Northern Hemisphere (30-90N) for all model simulations in the ssp585 scenario and corresponding historical simulation (scatter) and the linear/polynomial fit for the historical/scenario periods respectively (lines). Right panel: detrending for the ssp585 scenario simulation with EC-Earth3, considering different areas for the average.

L132-134. See my main comments. Why do the authors only use 4 EOFs for around 50% of explained variance?

- We added a clarification in the text at line 140, mentioning the sensitivity tests done in Fabiano et al. (2020a): "*Sensitivity tests performed in Fabiano et al. (2020a) for the EAT sector show that the changes in the regime patterns when considering for example 10 EOFs instead of 4 are negligible.*"

L136. I'm not sure that it is a very good justification, but OK. . .

- We are not interested here in assessing the "right" number of clusters to be used for the two sectors and we acknowledge at line 74 (original manuscript) that this number is still a matter of debate. We then adopt the most common choices in the literature, which are 4 clusters for both the EAT (Michelangeli et al., 1995; Cassou, 2008; Dawson et al., 2012; Madonna et al., 2017; Strommen et al., 2019; Fabiano et al., 2020) and PNA sectors (Straus et al., 2007; Weisheimer et al., 2014), in order to set up a framework to discuss future changes in the circulation.

L142. How is the centroid exactly defined: average, median, real day closest to the average etc?

- The centroid is defined in phase space as the average of all days (PCs) assigned to a certain cluster. This has been now clarified in the revised text at line 150: "[…] *we obtain a set of 4 cluster centroids, defined as the average of all PCs assigned to a certain cluster.*"

L145. Are all days classified and why? In some studies, the transition days are not classified, for example.

- Yes, we classify all days here. Although not classifying all days may be a legitimate approach, it requires the definition of a rule for excluding some of the days (threshold on the deviation from the mean state, on the velocity in phase space, ecc.) which makes the methodology more arbitrary. We judged it more conservative not to exclude any day from the clustering.

L146. Computed regimes: how are the computed regimes associated with the observed regimes? Is the associated observed regime the one with the stronger spatial correlation? Are there ambiguities? (e.g. a computed regime that looks like something intermediary between two observed regimes)

- Thank you for the comment. The matching of the computed and observed regimes is done minimizing the average RMS between all regime couples. This usually coincides with the best matching obtained maximizing the spatial correlation. However, as you point out, it may happen that some ambiguities arise when a computed regime is very far from the observed ones: this usually happens to two regimes at a time, which turn out to be a mixture of the two observed regimes they should have reproduced. This is quite rare, but when it happens, the corresponding metric in the Taylor plot is poor: the outliers in Figure 2 (with pattern correlation close to zero) are probably examples of this. The potential instability of the computed regimes is one of the reasons why we decided to use the projected regimes when considering the changes in the future scenarios.

L176-177. Not clear to me.

- These estimates on the variability on the 50-yr window are the standard deviation of the mean of the observed regime frequency and persistence in individual seasons (so the standard deviation divided by the square root of n_season - 1). This applies if we assume the consequent seasons to be independent. The actual variability on 50-yr windows might be larger than this due to, for example, decadal fluctuations in the WRs.

We rephrased the sentence in the text to explain this at line 186: "*The variability on a 50-yr window has been estimated as the standard deviation of the mean ($\sigma / \sqrt{n-1}$) of the seasonal frequency and persistence as 1.6% and 0.3 days respectively. However, the actual variability on these scales might be larger than this due to decadal basin-wide fluctuations.*"

L223-22. I'm not sure to understand the reasoning.

- We agree that the sentence has not been formulated well. Actually, our hypothesis here is that the tropically-induced modulation of the North Pacific regimes might be too strong in models. Therefore the regime structure turns out to be too "deep" and there is less room for larger deviations from the attractors. Molteni et al. (2020) showed that the response of the NAO index to the tropical Pacific forcing is well represented in models, while the teleconnection of the Atlantic sector with the Indian Ocean is not well caught. However, this does not really help our argument here. We changed the sentence in the revised manuscript as follows (lines 237-239): "*It is worth noting that – opposite to the EAT sector – models tend to produce larger variance ratios for the PAC regimes than it is observed, which might be due to an excess in the tropically-induced modulation of the PAC regimes in models.*"

L263. "Further analysis. . ." I understand, but it is unfortunate. It is a very interesting (and strange) results. Discussing some hypotheses would have been nice. Note that it may be relevant regarding the discussion about the "drivers", as the "drivers" could be implicated in these variations (if they really are "drivers").

- The most promising hypothesis is related to the aerosol forcing, that could have a role in driving in-phase AMV oscillations in the model simulations. This has been hypothesized for the observed AMV (see e.g. Zhang et al. Have aerosols caused the observed Atlantic Multidecadal Variability? J. Atmos. Sci. 70, 1135–1144 (2013); Qin et al. 2020, DOI: 10.1126/sciadv.abb0425). In turn, the AMV perturbs the observed frequency of the NAO+/- regimes (a positive AMV increases the NAO- frequency). It is not clear whether a similar process might be at work for the future scenario period, but the way seems promising.

We added a comment on this at lines 283-285: *"These might be related to the aerosol forcing, which has been hypothesized to have driven the observed AMV oscillations (Zhang et al., 2013; Qin et al., 2020). However, it is not clear whether a similar process might be at work for the future scenario period and further analysis on this topic will be carried out in a different study."*

L315. I think the mean changes in geopotential height at 500 hPa (maps) should be shown somewhere, maybe as the first figure of the paper.

- Thank you for the suggestion. The changes in the geopotential height at 500 hPa are dominated by the global positive trend, shown in Figure 3 of this response. This is the only component of the change in the geopotential height that we are not showing in the paper, but a corresponding figure has been added to the Supplementary material (Figure S11, left panel). The residual zonal and local trends of zg500 for ssp585 are shown in Figures 9 and 10. We judged these residual trends more interesting from a dynamical point of view, since they reflect the changes in the circulation that we observe.

L320 and legend of figure 8. It is not clear what the "trend" is here. In section 2.2.1 the trend is defined as filtered area-weighted average Northern Hemisphere geopotential height, and used to detrend the local geopotential height. Therefore, if my understanding is correct, here the authors look at the zonal mean "trend of detrended" data, right? It is not clear, and bit awkward from a vocabulary point of view. Please indicate the period used to compute the trends in the legend.

- Thank you for the comment. Yes, this is correct. The quantity shown differs from the zonal trend of the original geopotential height fields only by a constant, represented by the global NH trend. The complication here is that we used a polynomial detrending for the scenario data, so that's not simply a linear term. The period is the full scenario period 2015-2100, this is indicated at line 355.

L323: zonal mean trend or zonal mean trend anomaly, as said in the legend?

- We refer here to the zonal mean trend anomaly, the text was corrected.

L351. For the predictors I suppose that you also use the trends, as for the predictands (it is not said explicitely)? What is the period on which the trends are calculated for the predictors and the predictands? 2015-2100 I guess, but it is not explicitly said I think (also in the legend of Fig 10).

- Yes, we also use the trends for the predictors, divided by the global mean surface temperature trend. Trends are calculated on the 2015-2100 period for ssp585 and on 2006-2100 for rcp85. We added this information to the main text at line 388.

Section 4.2. Are there correlations between changes in EAT and PNA regime frequencies? I think knowing that might be useful for the discussion in this section.

- Thank you for the suggestion. We computed the correlations between the frequency trends for ssp585 in the two sectors. No significant correlation at the 95% level was found. The largest correlations are: AR and BR (+0.3), NAO- and PNA- (-0.3), SBL and PT (-0.3), SBL and PNA- (+0.35).

L352. Why 2 or 3? Please justify.

- The scope of Section 4.2 is to find the most significant set of potential drivers for the two sectors. In this regard, a lower number of predictors is desirable. We were inspired by Peings et al. (2018) and Oudar et al. (2020), that consider 2 and 3 drivers respectively, to search for the best 2 and 3 indices models for the two sectors.

Are (i) all the models with the ssp585 scenarios used, or, at it seems to be the case based on Table S1 (ii) only the models with the 4 scenarios are used even for this analysis? I think (i) would be much better as it would lead to a larger sample of models, which is quite small with (ii). It is even truer since some of the models are nearly duplicates: models at different resolutions, ESM and AOGCM from the same group etc., which decreases the "effective" sample size.

- The models with available daily zg dataset for ssp585 were 22 at the time when the analysis was done. This number is reduced to 19 with the constraint on the availability of all ssps, which were used for all analysis in the paper, including Section 4.2. However we also added here the 19 models from RCP8.5, to enlarge the sample to 38 models.

We understand that the number of models is somehow limited, nevertheless it can give some indication in terms of correlated quantities.

L362. I don't see figure S7. Are these correlations significant (with the issue of effective sample

size mentioned above it is difficult, or impossible, to do the test right, but it is still an interesting indication).

- Thank you for pointing this out, and sorry for forgetting to add the figure to the Supplementary, which is now included as Figure S12. The significance of the correlations at 99% and 95% level is indicated by the big and small white circles in Figure 11. The issue of the effective sample size might effectively reduce the significance of these correlations, we added a brief comment on this in the revised manuscript at line 404: "*However, the effective size of the sample may be smaller than the 38 models considered here (19 for both SSP5-8.5 and RCP8.5), since some models are closely related to each other and this might lower the significance of the regressions.*"

L387. Any idea of the reason(s) that might explain the improvements in CMIP6? Please discuss.

- CMIP6 models have been shown to improve in many regards with respect with the previous CMIP generation. This has been observed also for the northern mid-latitude circulation. In particular, models have been shown to have smaller biases in the blocking frequency (Davini and D'Andrea, 2020) and in the representation of storm-tracks (Harvey et al., 2020, https://doi.org/10.1029/2020JD032701). This finding is consistent with the improvement observed for the weather regimes. A possible reason for the better performance of the CMIP6 models might lie in the refined horizontal resolutions, which are significantly larger in CMIP6 models compared to CMIP5. In this regard, in Fabiano et al. (2020a) the models' response to increased resolution has been analysed in terms of the simulation of weather regimes and an overall improvement in the representation of   regime patterns and variance ratio was found.

[revised manuscript text omitted]